# *What Time Tells Us?* An Explorative Study of Time Awareness Learned from Static Images

**Dongheng Lin**[*]                                                                 *d.lin.2@bham.ac.uk*
*The MIx Group, University of Birmingham*

**Han Hu**[*]                                                                 *hxh347@student.bham.ac.uk*
*The MIx Group, University of Birmingham*

**Jianbo Jiao**                                                                 *j.jiao@bham.ac.uk*
*The MIx Group, University of Birmingham*

**Reviewed on OpenReview:** *https://openreview.net/forum?id=f1MYOG4iDG*

## Abstract

Time becomes visible through illumination changes in what we see. Inspired by this, in this paper we explore the potential to learn time awareness from static images, trying to answer: *what time tells us?* To this end, we first introduce a Time-Oriented Collection (TOC) dataset, which contains 130,906 images with reliable timestamps. Leveraging this dataset, we propose a Time-Image Contrastive Learning (TICL) approach to jointly model timestamps and related visual representations through cross-modal contrastive learning. We found that the proposed TICL, 1) not only achieves state-of-the-art performance on the timestamp estimation task, over various benchmark metrics, 2) but also, interestingly, though only seeing static images, the time-aware embeddings learned from TICL show strong capability in several time-aware downstream tasks such as time-based image retrieval, video scene classification, and time-aware image editing. Our findings suggest that time-related visual cues can be learned from static images and are beneficial for various vision tasks, laying a foundation for future research on understanding time-related visual context. Project page: `https://rathgrith.github.io/timetells_release/`.

> "Time is the moving image of eternity."
>
> *Plato*

## 1 Introduction

On our planet, the day-night cycle occurs every 24 hours, a phenomenon recorded systematically by various clock systems developed by human society. Surprisingly, such clock systems emerged much earlier than our recognition of Earth as a "blue marble" engaged in constant orbital movement (Dohrn-van Rossum, 1996). Although most people possess a vague, intuitive sense of current time (Moore, 1992), the origin of this metaphysical consciousness of time, which is a key concept for both our bodies and society, remains elusive. Research in neuroscience has revealed that visual stimuli from photoreceptors are crucial for the adaptation of mammals to day-night rhythms (Duffy & Czeisler, 2009). This implies that the concept of time for humankind could emerge from various visual experiences. Given the implicit relations between clock time and visual experiences, we are interested in asking: *1) Can neural networks gain similar awareness to clock time from solely visual stimuli* i.e. *static images? 2) If so, what implications does such time awareness tell us towards understanding the world?*

---

[*]These authors contributed equally.

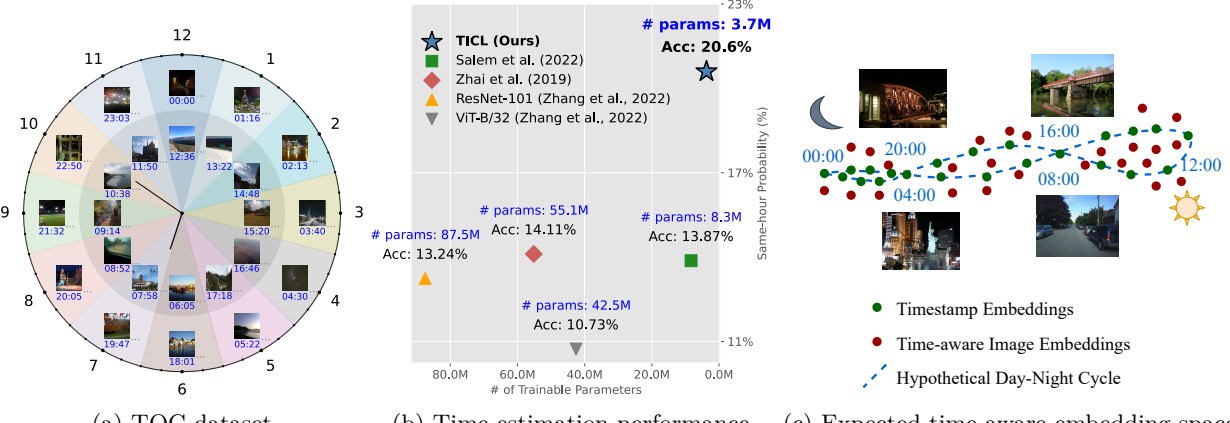

(a) TOC dataset      (b) Time-estimation performance      (c) Expected time aware embedding space

Figure 1: **An overview of our study,** in which we presented a new high-quality dataset for time-of-day estimation (a), based on which we propose a new approach, achieving state-of-the-art performance (b). We further explore the implications of learned time-aware embeddings (c), showing effectiveness over several time-related downstream tasks.

To answer these questions, in this study, we propose an approach to learn and disentangle the visual cues related to time from static images, via a pre-text task estimating the clock timestamps from images, and exploration on various downstream tasks to find their visual implications.

Learning to model captured timestamps of images requires a reliable natural image dataset with timestamps. There are previous surveillance camera datasets with fixed views, such as the Time of Year Dataset (TYD) (Volokitin et al., 2016) and other subsets of the Archive of Many Outdoor Scenes (AMOS) (Jacobs et al., 2009), featuring images captured by a few stationary webcams at different times of the day. However, these datasets do not reflect the complexity and diversity of views in real-world applications. To address this issue, Salem et al. (2020) proposed a mixed subset of AMOS and YFCC100M (Thomee et al., 2016), containing diverse samples. However, many images in this dataset suffer from incorrect timestamps due to unsynchronised time zones (Padilha et al., 2022), which undermines its reliability for learning robust time-awareness.

In addition to the challenge of lacking reliable datasets, designing effective solutions for the pre-text task also faces significant difficulty. Providing accurate clock time estimates requires the model to go beyond understanding basic illumination, as the task is complicated by inherent ambiguities between the clock timestamp and images. These ambiguities arise because daylight time is influenced by additional metadata, such as regional climate and seasonal variations that affect the duration of daylight hours (Volokitin et al., 2016; Sharma et al., 2016; Zhang et al., 2022). To cope with this issue, Salem et al. (2022); Zhai et al. (2019) introduced additional metadata inputs, aiming to model the joint conditional probabilities between geolocation, hour and date to provide performance improvements to the estimation task. While these works made reasonable and valuable improvements, they have introduced extra dependencies on additional metadata, limiting the generalisation ability when such metadata is unavailable as reported. On the other hand, they primarily focus on the specific task of clock time estimation, without exploring further implications of time to other applications. Whereas in this work, in addition to estimating more accurate timestamps, we further utilise the learned time and time-aware image features to investigate their impact on several other downstream tasks.

Specifically, due to the lack of high-quality data, we first curate a new benchmark dataset comprising social media images featuring diverse views and objects, along with manually verified reliable timestamps. Such a dataset has the potential to become the new de facto choice for future research. Secondly, we propose a Time-Image Contrastive Learning (TICL) approach that extracts time-of-day awareness from rich semantics from foundation vision-language model via contrastive learning outperforms all existing methods on the pre-text timestamp estimation task. Moreover, we conduct explorations of utilising such time-awareness

on several downstream tasks, including time-based image retrieval, video scene recognition, and time-aware visual editing, showing the indirect relations between time and scene understanding.

Note that this work is not aiming at purely time estimation, but more about an exploration of what the learned embedding tells us, through such a pre-text task. Our key contributions can be summarised as follows:

- We introduce Time-Oriented Collection (TOC), a new benchmark dataset containing 130,906 images with reliable timestamps (examples shown in Fig. 1a).

- We propose TICL, an approach jointly modelling time and related visual representations, achieving state-of-the-art (SOTA) performance on timestamps estimation from static images. Fig. 1b shows the achieved performance, boosting SOTA from **14.11%** to **20.6%**, while keeping small number of trainable parameters.

- We study the potential of the learned time-aware visual embeddings (Fig. 1c) by validating them on several downstream tasks (*e.g.* time-based image retrieval, video scene classification, and time-aware image editing), showing clear evidence of their effectiveness.

## 2 Related Works

### 2.1 Image datasets with timestamps

Estimating the time of day from static images is a notable and underexplored challenge. Earlier studies were hampered by the scarcity of datasets with images paired with accurate local timestamps. Many images from social networks often have metadata that is inaccurate, missing, or uncalibrated to local timezones. To cope with this, some researchers have turned to webcam image datasets, which naturally include accurate timestamps. However, these datasets are limited to fixed views and are often degraded by noise, low light, or obstructions, hindering their generalisation to diverse applications.

For example, established social media image datasets such as MIRFLICKR-1M (Huiskes & Lew, 2008) and YFCC100M (Thomee et al., 2016) were found to contain many unnatural non-photographic images (*e.g.* memes, scribbles) and inaccurate timestamps due to unsynchronised clocks and other sources of inconsistency. On the other hand, webcam datasets contain only fixed stationary views, such as AMOS (Jacobs et al., 2007) and TYD dataset (Volokitin et al., 2016), which fail to represent the complexities of temporal variations within diverse environments. The CVT-Time dataset (Salem et al., 2020), despite combining stationary webcam images with YFCC100M subsets with images captured by smartphone, still struggles with unreliable timestamps and low-quality webcam images.

### 2.2 Clock timestamp estimation

Previous work have studied joint attribute estimation of images, including captured clock time, date, and geolocation. *In this work, we focus solely on clock time estimation regardless of other fields in timestamps (e.g., date, year),* since we are primarily interested in relations between clock time itself originated from human activities (Moore, 1992) to visual cues. Volokitin et al. (2016) used VGG-16 to classify temperature, month, and hour intervals from images taken by 6 webcams during daylight, which is insufficient for comprehensive day-long analysis. In addition to such earlier simple approaches, Zhai et al. (2019) worked with a mixed dataset of Flickr and webcam images, classifying images taken at the same hour but in different months into 288 classes, optionally incorporating geolocation inputs. Similarly, Salem et al. (2022) used webcam images, predicting month, week, and hour as dependent tasks trained jointly while considering geolocations as optional inputs. Such joint predictions improve hour-based time-of-day classification by leveraging metadata cues on regions and climate, which correlate with daylight length. However, such dependency also puts risks on generalization ability when there are no reliable geolocation or date metadata available (Salem et al., 2020; Zhai et al., 2019). Therefore, we deliberately chose to use only input images for clock time prediction, without utilising any additional metadata with regard to the generalization problem acknowledged in previous work.

## 3 Time-Image Contrastive Learning

Before introducing the proposed method, we revisit the problem formulation and goal of our study. Unlike previous timestamp-prediction architectures that simply attach a classifier to an image backbone (Salem et al., 2020; Zhai et al., 2019; Volokitin et al., 2016), our goal is *not* to engineer another bespoke head. Instead, we introduce a minimal, generic recipe that transforms a certain environmental attribute (in our case, the clock time) into *embeddings* that elucidates the implications of world understanding from the attribute. The novelty of TICL therefore lies in providing explicit semantic-aware time embeddings for further investigations with its robustness on time encoding validated by superior pre-text time estimation task performance.

For the clock-time estimation of images specifically, we seek to train a model $f_\theta(\cdot)$, which predicts a timestamp $\hat{t}$ given input image $x$. The estimate can be written as $\hat{t} = f_\theta(x)$. While regression seems ideal owing to the continuous nature of time, it faces significant challenges. The cyclic nature of the clock introduces discontinuity to regression methods that treat target values as scalars within a range that is a disconnected set (Zhou et al., 2019). In regression, cyclic data often causes $\hat{t}$ to cluster near the midpoint of the range (Adams & Vamplew, 1998). For instance, timestamps like 23:59 and 00:00, despite their visual similarity, are treated as opposite extremes on the time scale. In such cases, the regression model tends to reach a sub-optimal solution around 12:00, which is far from accurate. Apart from this extremal case, the sensitivity of the scalar time values also encourages the model to predict an average ground-truth value across visually similar images. Encoding time into cyclic space partially mitigates scalar discontinuities (Adams & Vamplew, 1998; Kazemi et al., 2019), but sensitivity issues still limit performance (see detailed analysis in Appendix C).

This justified why prior studies have employed classification over discrete time periods (*e.g.* hours), in which $\hat{t}$ has finite value options corresponding to classes. Classification mitigates the above issue in regression by simplifying the model to give a coarser estimate from mutually orthogonal discrete classes. Even for boundary cases like 23:59 and 00:00, the classification model tends to predict one of the adjacent classes (*e.g.* 23:00 or 00:00), which is more reasonable. However, the orthogonality of one-hot vectors (Rodríguez et al., 2018) overlooks the relationships (partial order, cyclic) between time periods.

On top of these observations, we propose *Time-Image Contrastive Learning* (TICL), a multi-modal approach that jointly learns time and image representations via cross-modal contrastive learning, inspired by GeoCLIP (Vivanco et al., 2023). Each input image $x_i$ is associated with a label $t_i \in \mathbb{R}^C$ indicating its time period. Empirically, we fix $C = 24$ for all the results in the main paper for a fair comparison with previous work (see further discussions on the choice of $C$ in Appendix C.2). Each one-hot vector $t_i$ is projected into a high-dimensional representation space $\mathbb{R}^K$ using a *Time Encoder* $T_i = f_{\theta_T}(\cdot)$, where $K = 768$ to match the dimensionality of the image representation.

As visualised in Fig. 2, during training we maximise the cosine similarity between the CLIP image feature $I_i = f_{\theta_I}(x_i)$ and its corresponding time-class embedding $T_i = f_{\theta_T}(t_i)$. Here, $f_{\theta_I}(\cdot)$ denotes the combined operation of the frozen CLIP image encoder and an *Image-Time Adaptor* (ITA). The alignment is optimised by minimising the contrastive loss (He et al., 2019), as defined in Eq. (1), where $\tau$ is a learnable temperature that controls the sharpness of the softmax distribution (Wu et al., 2018). At inference, TICL flexibly supports both classification at any class granularity and nearest-neighbour inference (see Appendix B.1, Appendix C.2 and Appendix B.2).

$$\mathcal{L}_B = -\sum_{i=0}^{B-1} \log \frac{\exp(I_i \cdot T_i / \tau)}{\sum_{j=0}^{B-1} \exp(I_i \cdot T_j / \tau)} \tag{1}$$

Several key intuitions support this design. Previous work has shown that combining additional geolocation and date information can improve the performance of time estimation, but reliance on such attributes propagates errors from prior to posterior predictions (Salem et al., 2020). We observed that the CLIP image encoder is a powerful foundation model capturing rich semantic context, exhibiting strong zero-shot capabilities in geo-localisation and scene recognition (Radford et al., 2021; Agarwal et al., 2021; Vivanco et al., 2023). These results suggest that CLIP implicitly encodes cues (*e.g.* climate, region) relevant to

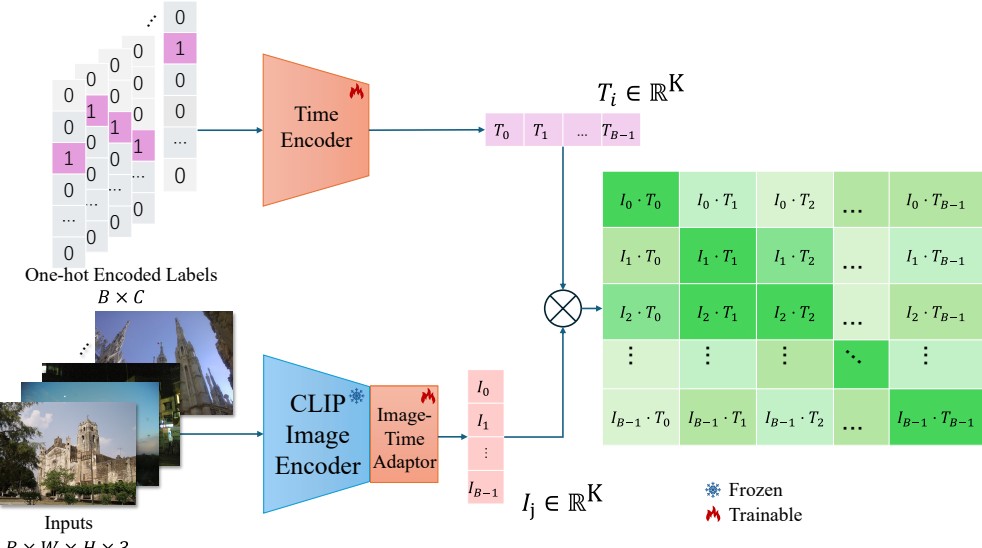

Figure 2: **Overview of TICL.** Given static images and one-hot time labels, two encoders (Time Encoder and image encoder + ITA) project inputs into a shared feature space; a contrastive loss aligns the corresponding pairs.

Table 1: Classification accuracy on the TOC test set of the baseline model Salem et al. (2022) when using different training datasets.

| Training Dataset [†] | Top-1 ↑ | Top-3 ↑ | Top-5 ↑ |
|---|---|---|---|
| Original Salem et al. (2020) | 12.02% | 34.05% | 56.45% |
| **Cleaned TOC (ours)**[‡] | **13.87%** | **39.36%** | **60.71%** |

[†] In training datasets, overlapped samples from the test set are excluded.
[‡] The new dataset details are introduced in Section 4 and Appendix A

timestamp estimation. Therefore, we *freeze* the CLIP backbone and extract these cues directly, rather than ingesting raw geolocations or season inputs that may themselves be noisy (Salem et al., 2022).

Another benefit comes from the learnable time embedding in the contrastive scheme. In a vanilla classifier, the final output $\hat{y}$ is confined to the simplex $\{\|\hat{y}\|_1 = 1, \hat{y} \in \mathbb{R}^C\}$, where each target is a fixed one-hot vector. Samples are optimised only toward their own target, and activations to related classes may be suppressed (He & Garcia, 2009). In contrast, our method endows each time class with a *trainable* vector that absorbs shared information among temporally adjacent samples, aligning timestamps and visual inputs more effectively—especially for tail classes. These learnable embeddings also prove useful in downstream tasks (Section 6).

In summary, we expect TICL combines the benefit from the orthogonality of one-hot encoded labels and flexibility of learnable high-dimensional embeddings. This simple yet principled design delivers consistent gains over prior estimators and alternative time-encoding schemes (Rahimi & Recht, 2007; Kazemi et al., 2019; Salem et al., 2022; Zhai et al., 2019) (see Sections 5.2 and 5.3), while remaining computationally lightweight and conceptually generalisable.

# 4 Benchmark Dataset TOC

With regards to problems Section 2.1, we introduce a new benchmark Time-Oriented Collection (TOC) dataset consisting of high-quality images sourced from social media, featuring reliable image metadata. We collected 117,815 training samples and 13,091 test samples from the Cross-View Time (CVT) (Salem et al.,

2020), mitigating various limitations in previous datasets. This dataset reflects real-world scenarios and human activities, making time-of-day estimation more applicable to potential practical applications.

During dataset curation, we manually filtered out unnatural, non-photographic images from the CVT dataset and calibrated the timestamps to match the images. To accelerate the process, we utilized ResNet18 features of the images to quickly identify the outliers in deep image feature space for different periods of the day. After which, we conducted meticulous manual inspection for each outlier image to check if their timestamp or contents are natural and valid (see more details in Appendix A). This revised dataset reflects natural variations in human activity throughout the day, with improved reliability in terms of time metadata. As evidence, Table 1 shows a performance gap on the same test set using different levels of filtering on the CVT dataset, justifying the effectiveness of the filtering process indicated by improvements in baseline model performance, suggesting that repetitive surveillance-camera-sourced and unnatural non-photographic images we removed do not help the model in better time recognition for images in the wild. A few examples of the exact format and appearances of the remaining samples within the TOC dataset are provided in Fig. 3. Geolocation distribution of the sample images within our final TOC dataset in Fig. 3 also suggests that, due to the inherent geographic distribution of the internet, the northern hemisphere has more data captured by nature. Our dataset well represents such natural distribution. Further information and statistics about the dataset are available in Appendix A.

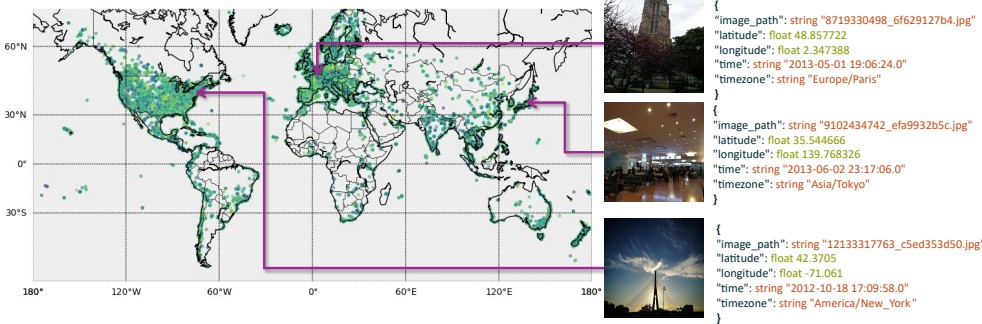

Figure 3: **Sample images and metadata from the TOC dataset w.r.t. GPS coordinates.** Metadata contains several fields indicating timestamps and geolocations. The samples spread across all the continents and show a natural distribution of internet images, where the southern hemisphere has relatively fewer samples due to a sparser population of photo capturing.

# 5 Experiments

## 5.1 Experiment Setting

**Dataset and metrics:** We use different evaluation metrics to measure performance on image clock time estimation tasks: top-k classification accuracy with $k = 1, 3, 5$, and Time Mean Absolute Error (MAE) on a minute basis. In addition to the TOC test set, to better evaluate the generalisation ability of the proposed method, we selected a subset of the AMOS dataset (Jacobs et al., 2007) as an additional test set. This additional test set contains 3,556 images with high SNR (which ensures good sample quality) captured by 53 stationary surveillance cameras with a more balanced time label distribution (see curation process and statistics in Appendix A.2). That is, all the compared models are trained solely on the TOC training set and evaluated on different test sets to demonstrate generalisation ability across different domains.

**Implementation details:** For our proposed TICL, we use Adam optimiser with an initial learning rate of $5 \times 10^{-4}$ and a weight decay of $1 \times 10^{-6}$. The training process spans 20 epochs, with the learning rate halved every 2 epochs and a batch size of 512. The temperature parameter is initialized to 0.07. All input images are resized to $224 \times 224$. For a fair comparison, we retrained all the previous baseline methods on the cleaned TOC train set, using the best training configurations reported in Zhang et al. (2022); Zhai et al. (2019); Salem et al. (2022) respectively. Additional details about implementations are available in Appendix B.1.

Table 2: Time estimation performance on our TOC dataset and the AMOS test set.

| | TOC test set | | | | AMOS test set[†] | | | |
|---|---|---|---|---|---|---|---|---|
| | Top-1 acc ↑ | Top-3 acc ↑ | Top-5 acc ↑ | Time MAE (min.) ↓ | Top-1 acc | Top-3 acc | Top-5 acc | Time MAE (min.) |
| BLIP-VQA-base (Li et al., 2022) ¶ | 9.36% | - | - | 241.58 | 7.28% | - | - | 302.82 |
| GPT-4o-mini (OpenAI et al., 2024) ¶ | 15.35% | - | - | 161.39 | 11.15% | - | - | 216.83 |
| Zhang et al. (2022) (ResNet-101) | 13.24% | 37.30% | 58.23% | 177.84 | 7.85% | 24.26% | 40.10% | 261.89 |
| Zhang et al. (2022) (ViT-B/32) | 10.73% | 31.21% | 49.05% | 195.33 | 7.25% | 21.03% | 32.93% | 263.87 |
| Zhai et al. (2019) | 14.11% | 40.47% | 65.94% | 188.78 | 9.14% | 27.95% | 45.36% | 262.68 |
| Salem et al. (2022) | 13.87% | 39.36% | 60.71% | 186.44 | 8.63% | 26.49% | 42.58% | 255.20 |
| **TICL (Ours)** | 20.60% | 49.01% | **67.82%** | 171.65 | **13.55%** | **38.49%** | **57.28%** | **187.87** |
| **TICL-Nearest-Neighbour (Ours)[‡]** | **25.67%** | **49.32%** | 66.74% | **156.24** | 11.14% | 31.01% | 48.84% | 220.94 |
| Zhai et al. (2019)[§] | 15.01% | 42.54% | 68.24% | 185.34 | 8.85% | 24.12% | 38.63% | 268.41 |
| Salem et al. (2022)[§] | 13.53% | 38.47% | 59.10% | 176.70 | 8.16% | 23.88% | 39.67% | 257.00 |

[†] Experiments on this test set are conducted in a zero-shot manner, in which we directly evaluate models trained solely on the TOC dataset.
[¶] The result is obtained via direct Visual Q&A on the Visual Language Model with details in Appendix B.2.
[‡] Results in this row are achieved via Nearest-Neighbour style inference. We choose the time labels of nearest neighbours from the train dataset as estimations (see details in Appendix B.2).
[§] These methods take additional known geolocation metadata inputs. Therefore, it's unfair to directly compare them with other methods. So we put them here just for reference.

## 5.2 Time estimation performance

As shown in Table 2, TICL not only outperforms all previous pure vision methods but also outperforms previous methods that require additional geolocation inputs on most metrics. TICL also demonstrates better performance in the additional AMOS test set, thereby indicating better generalisation ability. In summary, our experimental results indicate an overall improvement of the proposed methods in clock time estimation, especially in terms of accuracy and generalisation ability.

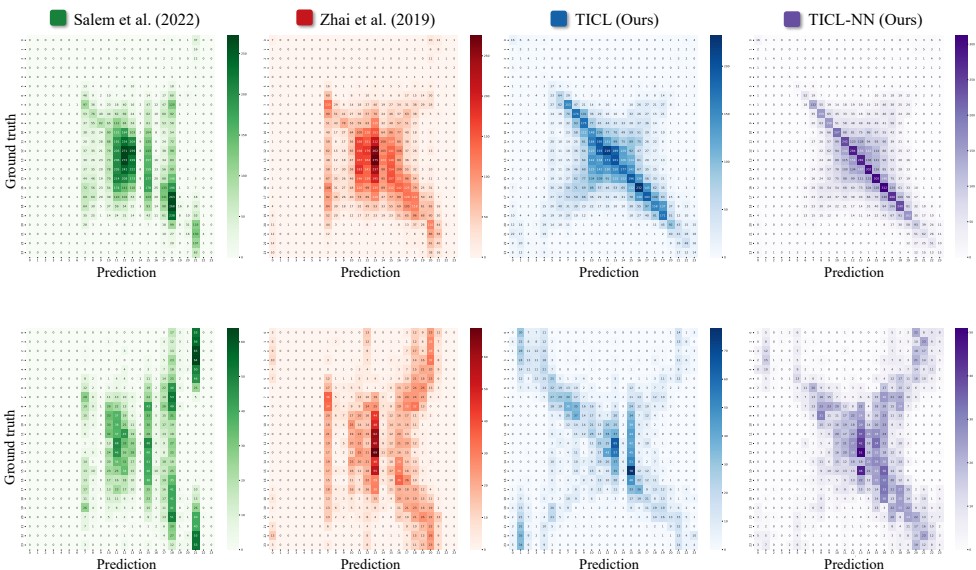

Figure 4: **Confusion matrices.** They provide more detailed comparisons throughout the 24 hours on our TOC test set (top), and the AMOS test set (bottom).

**Additional error analysis on pre-text tasks** In addition to the quantitative results, we also visualised the confusion matrices in Fig. 4 to provide a more in-depth evaluation of the task. An interesting finding is that both Salem et al. (2022) and Zhai et al. (2019) overlooked minority classes in the training set (classes from 1 a.m. to 5 a.m.), resulting in nearly no predictions for these classes on both test sets. This indicates a notable bias in these models towards classes during hours of intense human activity, when more images are present in dataset. In contrast, our proposed TICL method exhibits more balanced class-wise distributions of positive predictions on both test sets, suggesting better estimation fairness. The general trend in all the confusion matrices also suggests the remaining challenges faced by all methods. Notable anti-diagonal patterns indicate inherent visual ambiguities of the clock system w.r.t. appearances.

Table 3: Detailed component analysis of the proposed method design.

| Image Encoder[†] | $f_{\theta_T}$[‡] | $f_{\theta_{\mathrm{ITA}}}$[§] | TOC test set | | | | AMOS test set | | | |
|---|---|---|---|---|---|---|---|---|---|---|
| | | | Top-1 acc ↑ | Top-3 acc ↑ | Top-5 acc ↑ | Time MAE (min.) ↓ | Top-1 acc | Top-3 acc | Top-5 acc | Time MAE (min.) |
| DINOv2-base | ✗ | ✗[¶] | 7.69% | 23.36% | 38.61% | 302.84 | 5.65% | 17.12% | 27.28% | 319.09 |
| | Ours | ✗ | 8.01% | 23.84% | 39.06% | 295.34 | 5.23% | 17.35% | 29.22% | 320.76 |
| | ✗ | ✓ | 1.02% | 3.29% | 12.04% | 486.77 | 4.11% | 11.41% | 19.62% | 381.92 |
| | Ours | ✓ | 9.53% | 27.34% | 44.17% | 254.49 | 5.09% | 14.74% | 25.16% | 327.72 |
| SwinV2(B) | ✗ | ✗[¶] | 11.45% | 32.27% | 51.08% | 240.77 | 7.87% | 22.49% | 36.81% | 281.80 |
| | Ours | ✗ | 11.64% | 32.13% | 50.33% | 243.86 | 7.51% | 22.36% | 37.54% | 288.21 |
| | ✗ | ✓ | 12.74% | 33.65% | 52.06% | 222.76 | 6.75% | 23.76% | 38.41% | 284.30 |
| | Ours | ✓ | 13.37% | 34.94% | 52.93% | 216.17 | 7.37% | 22.98% | 38.08% | 276.66 |
| ConvNeXt(L) | ✗ | ✗[¶] | 11.59% | 32.93% | 50.88% | 240.64 | 6.41% | 21.68% | 37.63% | 300.74 |
| | Ours | ✗ | 11.86% | 32.81% | 50.18% | 240.80 | 6.10% | 20.66% | 35.85% | 302.45 |
| | ✗ | ✓ | 13.51% | 35.29% | 52.76% | 216.28 | 7.71% | 24.33% | 39.96% | 275.23 |
| | Ours | ✓ | 14.67% | 36.75% | 54.60% | 204.19 | 8.27% | 24.78% | 40.86% | 263.03 |
| CLIP (ViT-L/14) | ✗ | ✗[¶] | 16.66% | 44.43% | 65.07% | 193.66 | 12.37% | 36.95% | 55.96% | 200.93 |
| | Ours | ✗ | 16.73% | 44.05% | 63.99% | 195.41 | 13.50% | 38.49% | **58.30%** | 189.99 |
| | ✗ | ✓ | 18.60% | 46.41% | 65.98% | 181.22 | 12.57% | 37.51% | 57.23% | 189.69 |
| | ✗ | $f_{\theta_{\mathrm{ITA}}} \oplus f_{\theta_T}$ | 19.26% | 45.40% | 62.92% | 189.97 | 11.42% | 35.65% | 54.06% | 197.09 |
| | RFF | ✓ | 16.75% | 35.14% | 46.61% | 206.50 | 6.07% | 15.78% | 22.27% | 290.70 |
| | T2V | ✓ | 17.70% | 45.69% | 66.11% | 185.89 | 7.37% | 21.74% | 35.10% | 264.25 |
| | Ours | ✓ | **20.61%** | **49.01%** | **67.83%** | **171.65** | **13.55%** | **38.50%** | 57.28% | **187.87** |

[†] All image encoders are frozen feature extractors with pretrained features provided by corresponding PyTorch libraries (Wolf et al., 2020; maintainers & contributors, 2016).
[‡] $f_{\theta_T}$ denotes the Time Encoder module. When $f_{\theta_T}$ is absent, only one-hot encoding is used to represent the clock timestamp, and the outputs of $f_{\theta_I}$ need to be projected to 24 dimensions. RFF, T2V means that we uses off-the-shelf encoding methods for low-dimension/cyclic vectors from Rahimi & Recht (2007); Kazemi et al. (2019).
[§] $f_{\theta_{\mathrm{ITA}}}$ denotes the Image-Time Adaptor. When it is absent, only the backbone feature extractor and time encoder are used.
[¶] The baseline with neither of the $f_{\theta_T}$, $f_{\theta_{\mathrm{ITA}}}$ components simply has a linear layer after Image Encoder projecting the features to 24 dimensions.

## 5.3 Detailed component analysis

In this section, we present the ablation study investigating the effectiveness of each module in the proposed TICL model across different configurations. To ensure a fair comparison, we use a classification-based inference pipeline for all experiments (see implementation details in Appendix B.2). Table 3 provides performance comparisons under various settings, including different backbones (Tan & Le, 2021; Oquab et al., 2023; Liu et al., 2022a;b) within the image encoders.

**Impact of different backbone image encoders:** The differences in performance across the image encoder backbones highlight the effectiveness of the CLIP Image Encoder. Thanks to its rich semantic representations, the CLIP Image Encoder consistently achieves better results across all configurations than other backbones.

**Ablation on proposed modules:** We observed that the Time Encoder $f_{\theta_T}$ and the Image-Time Adaptor $f_{\theta_{\mathrm{ITA}}}$ have varying effects when used individually, either slightly improving or degrading the baseline. However, when both modules are employed simultaneously, they lead to universal improvements across all image encoder backbones, underscoring the joint contribution of the Time Encoder and Image-Time Adaptor.

**Ablation on different time encoding methods:** We also tested the performance using other variants of Time Encoder. RFF (Rahimi & Recht, 2007) encodes input (hour, minute) into 512-dim vectors to align with ITA outputs directly using the same dynamic queue as in Vivanco et al. (2023), which is outperformed by our methods on TOC test set and does not generalise well on AMOS test set. In addition, T2V (Kazemi et al., 2019) based Time Encoder also shows similar problems on its performances. These comparisons suggest that the one-hot class embeddings exhibit better generalisation ability and performance on most metrics. A possible explanation could be that, the sensitivity of accurate time encoding results in some clock timestamp embeddings being assigned with very limited training samples to represent them. This makes them not robust against the visual ambiguity of time, as images with the same clock time could have very different appearances due to variations in geolocation, season, and climate. In contrast, clock time class embeddings for each hour are vaguely associated with many samples that lie in the same hour interval. Representing target clock timestamp embeddings using spectrums of temporally close samples may reflect the ambiguity of clock time w.r.t. image appearances, making the estimates more robust and generalizable. (See more analysis in the Appendix C.1).

Table 4: **Joint localisation of geolocation and time.** probabilities that the top-1 retrieved image has GPS coordinates' L1-difference ≤ 0.01 and a clock time L1-difference ≤ 30 minutes to query images.

| Chance | Salem et al. (2022) | Zhai et al. (2019) | CLIP (ViT-L/14) | **TICL (Ours)** |
|---|---|---|---|---|
| 0.03% | 2.08% | 4.17% | 6.25% | **10.42%** |

## 6 What Time Tells Us on Downstream Tasks?

Apart from the possible media forensics application that time estimation can be applied to (Padilha et al., 2022). We also interested in the relation to other computer vision tasks and the capability of the learned time awareness. Therefore, in this section, we explore 1) time-based image retrieval which is a direct use of the model in retrieval & recommendation applications (Section 6.1), 2) video scene classification, revealing an interesting connection between static images and dynamic visual scenes learned from timestamp supervision (Section 6.2), and 3) time-based image editing, showing the model learned can provide proper perceptual guidance to the generative models (Section 6.3).

### 6.1 Time-based image retrieval

An intuitive application of the time-aware model is time-based image retrieval. It aims to effectively retrieve images from a database with a similar captured time of day to the query images. We consider a zero-shot vector search engine that retrieves the nearest neighbours of query images based on their time-aware feature similarities. To evaluate this task, we separated the TOC test set into 13,043 database images and 48 query images spanning all 24 hours. The performance is measured using Recall@k reported in Fig. 5. Images retrieved with a time difference of no more than 30 minutes from the queries are considered as positives. The results clearly show that the proposed TICL model achieves the best performance across all Recall@k metrics.

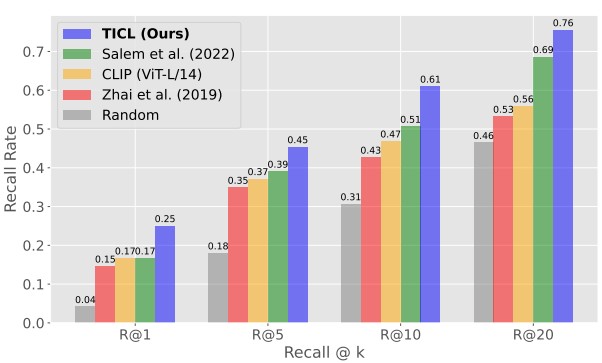

Figure 5: Recall@k for time-based image retrieval.

We also analysed the distribution of metadata differences between the retrieved images and their corresponding query images. Specifically, Fig. 6a illustrates the distribution of clock time errors among the top-100 retrieved samples for different features. The results show that TICL retrieves a higher percentage of images with smaller time errors compared to other features. Fig. 6b further shows the geolocation error distribution. Images retrieved by vanilla CLIP embeddings are geographically closest to the queries, suggesting that CLIP represents a rich understanding of scene priors strongly related to geolocations, which was delineated in some previous Visual Place Recognition (VPR) work using CLIP backbone (Radford et al., 2021; Keetha et al., 2024; Vivanco et al., 2023). We suspect that this contextual awareness is partly inherited by TICL, which achieved moderately better performance than other time-aware features of previous work. From this observation, we suspect that TICL disentangled time-aware features from other metadata attributes. To validate this hypothesis, for each query image, we consider an additional task of localising geolocation and time jointly using retrieval. As shown in Table 4, the advantage of TICL suggests it has a more balanced capability of understanding geolocation and time jointly than other models.

### 6.2 Video scene classification

Understanding dynamic scenes is an important challenging problem that visual models currently face (Miao et al., 2021). A fundamental task in this domain is video scene classification. Pretraining models on static images with object categories have been proven to be helpful in video classification (Carreira & Zisserman, 2017). Intuitively, dynamic scenes represented in videos have temporally consistent frames within.

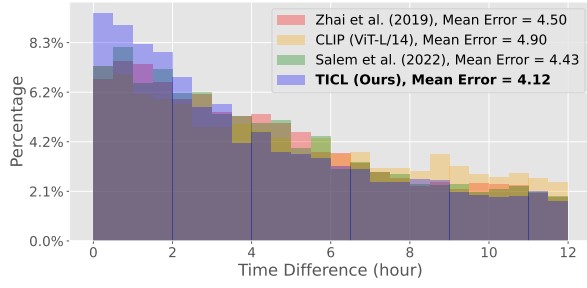
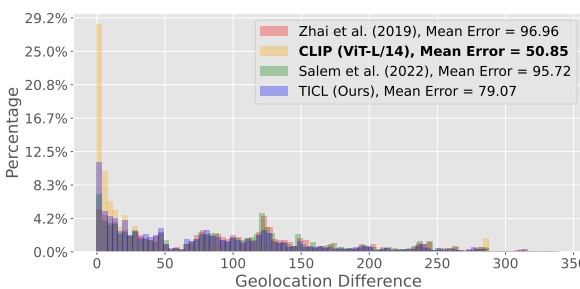

(a) Retrieval time error distribution

(b) Retrieval geolocation error distribution

Figure 6: **Comparison of geolocation and time error distribution.** It is collected among top-100 retrieved images using different feature extractors.

Table 5: Performances on the video scene classification task.

| Classifier | Hollywood2-Scene ↑ | YUP++ [†]↑ | 360+x (Panoramic) ↑ | 360+x (Third-view) ↑ |
|---|---|---|---|---|
| VideoMAE (Tong et al., 2022) | 48.83% | 97.29% | 53.70% | 54.55% |
| VideoMAE + CLIP (ViT-L/14) | 52.92% | **98.33%** | 57.40% | 50.91% |
| VideoMAE + Salem et al. (2022) | 45.53% | 97.50% | 44.45% | 52.72% |
| VideoMAE + Zhai et al. (2019) | 51.03% | 97.71% | 48.15% | 56.36% |
| **VideoMAE + TICL** | 56.53% | **98.33%** | **59.26%** | **58.18%** |
| CLIP (ViT-L/14) (Linear Probing) | 39.69% | 97.08% | 35.19% | 11.10% |
| **TICL (Linear Probing)** | **57.04%** | **98.33%** | 51.85% | 42.59% |

[†] We use an unofficial train/val/test split of 5:1:4, since the original 1:9 train/test split overfit prematurely.

Therefore, despite dynamic scene categories seeming to be irreverential to the time of day, we are particularly curious about whether the proposed TICL model, which is pretrained to estimate clock time for input static images, can provide additional understanding of a continuous sequence of frames other than discrete moments represented by static images.

**Experiment setup:** To assess whether our time-aware models provide valuable priors for understanding different categories of dynamic scenes, we provide classification results under two different constructions. 1) We concatenate the time-aware features from different frozen feature extractors to pretrained VideoMAE backbone (Tong et al., 2022), 2) directly run linear probing on video frames with the frozen feature extractors. We compared the performances under different feature extractors on various scene datasets including Hollywood2-Scene (Marszałek et al., 2009), YUP++ (Derpanis et al., 2012) and 360+x (Chen et al., 2024). Please refer to Appendix E for implementation details and other experimental settings.

**Possible correlations between the time of day and scene:** According to Table 5, TICL features provide consistent improvements to the scene classification task under different settings. The most straightforward explanation for this boost is that scene classes are correlated with the learned time of day by definition. To prove this, we visualized the cosine similarity between certain text embeddings of certain scenes that clock time class embeddings, as shown in Fig. 7. The imbalanced distributions proved the conceptual correlation of scenes to time due to human activity patterns.

**Consistency in time-aware frame embeddings:** As shown in Section 6.1, the TICL representations can capture similarities between images with close clock times. Natural videos, although they sometimes involve drastic subjects or view movement, frames within each should still represent continuous time periods. TICL features for frames across the whole video should be more consistent than those of vanilla CLIP, which have stronger locality per frame (Tang et al., 2021). This intra-video consistency allows for more general time-aware priors extracted using TICL. The t-SNE visualisation of the video features in Fig. 8 supports this claim, showing that TICL features are more separable than vanilla CLIP features (see Appendix E.1 for a more in-depth analysis of the phenomena and claims above).

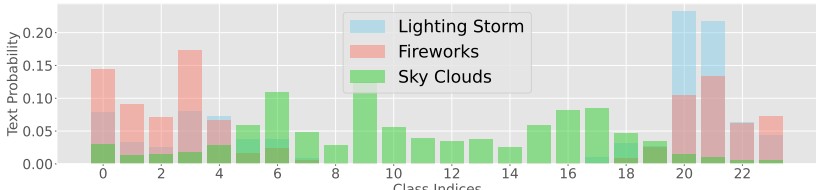

Figure 7: **The probability distribution for different input text queries of scenes that are seemingly irrelevant to time.** This is calculated by **Softmax** $= \frac{\exp(T_{CLIP} \cdot T_i)}{\sum_{j=0}^{|C|-1} \exp(T_{CLIP} \cdot T_j)}$ where $T_i, T_j, T_{CLIP}$ are the TICL clock time class embeddings and CLIP text embeddings.

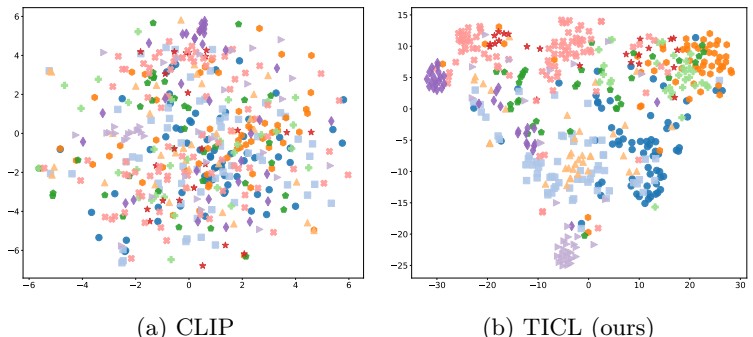

(a) CLIP           (b) TICL (ours)

Figure 8: **t-SNE visualisation comparison.** It compares video features before the final classifier layer using either (a) CLIP or (b) TICL, on the Hollywood2-Scene dataset (Marszałek et al., 2009), each different scatter point shape/colour corresponds with classes.

## 6.3 Time-aware image editing

As aforementioned in Section 3, the TICL model can provide the corresponding embeddings for certain periods of the day. Therefore, it is natural to consider using these clock timestamp embeddings as guidance to edit images toward different classes. To assess the extent to which clock time embeddings aid this task, we adopted the following experiment framework from Patashnik et al. (2021) that conducts image editing via latent vector searching through optimisation steps instead of tuning the models directly.

**Experiment setup:** To provide comprehensive evaluations, we conducted experiments on three different baseline StyleGAN2 models (Karras et al., 2020b) focusing on different subjects trained on (Skorokhodov et al., 2021; Yu et al., 2015). The pretrained generator weights are adopted from existing codebases Pinkney (2024); Epstein et al. (2022); Karras et al. (2020a). The editing pipelines were restricted to follow the same latent optimisation baseline method introduced in StyleCLIP (Patashnik et al., 2021). Additionally, we designed a new time-aware synergy loss combining directional CLIP loss and TICL feature similarity loss. Specifically, the editing process can be formulated as:

$$\underset{w \in \mathcal{W}^+}{\arg\min} \left( \lambda_1 \mathcal{L}_{TICL} + \lambda_2 \mathcal{L}_{\text{CLIPdir}} + \lambda_{l2} \|w - w_{\text{source}}\|_2 \right)$$

in which $w, w_{source}$ represents latent vectors for ongoing edit outcomes and original images, (design, hyperparameter and implementation details in Appendix F.1).

**Qualitative evaluation:** The proposed time-aware synergy loss yields the most plausible synthesis outcome as illustrated in Fig. 9. The limitations of solely text-guided image editing methods could be due to their susceptibility to certain adversarial solutions fooling CLIP image encoders with certain patterns only (Liu et al., 2021). Specifically, Fig. 9 shows the vanilla StyleCLIP edits using the CLIP loss tend to focus on the general tint of the image but fail to reflect realistic illuminations. We find that replacing the CLIP

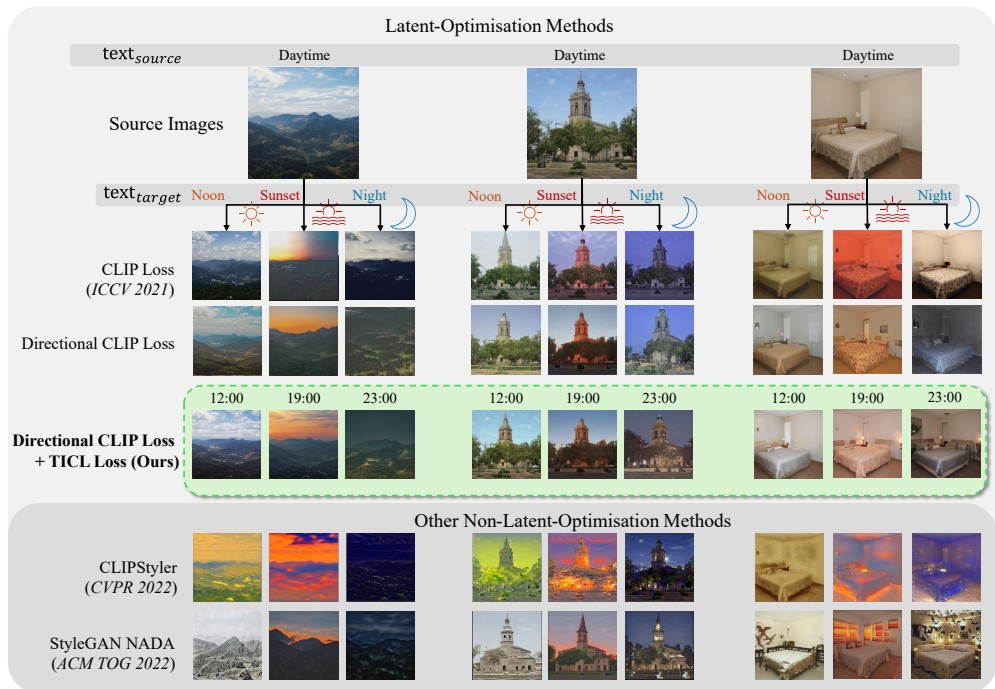

Figure 9: **Time-aware image editing.** It shows the results of applying our time-aware editing method (green overlay) on three different StyleGAN2 models trained on LHQ-Landscape (Skorokhodov et al., 2021), LSUN-Church, and LSUN-Bedroom (Yu et al., 2015) datasets. The results of other non-latent optimisation methods are also demonstrated (under grey overlay).

loss with a directional variant introduced in previous work (Gal et al., 2021; Kwon & Ye, 2022) can assist in overcoming larger domain gaps. Despite showing improvements over the baseline editing method, the results still show unrealistic artefacts and shape distortions. These limitations show the necessity of incorporating additional time-aware features other than just guidance text embeddings when computing loss functions for image edits. Our qualitative evaluations demonstrated the effectiveness of the TICL embeddings on the specific task. We also included other baseline method results that work under different frameworks other than latent optimisation for a more comprehensive comparison. See more quantitative evaluations (Table 12), user studies (Table 13) and results on TICL-aided editing with diffusion models in Appendix F.1 and Appendix F.2 respectively.

## 7    Conclusion

In this paper, we tried to answer the question of *what time tells us*, through exploring the pretext task of time-of-day estimation and downstream tasks. A new reliable benchmark dataset, *TOC* was introduced to support the pretext task, consisting of images captured in natural settings with verified timestamps. This dataset addresses the limitations of existing datasets by providing a more diverse and realistic collection of images that better reflect daily visual experiences. Building upon that, a new learning paradigm (*TICL*) was proposed, which aligns clock timestamp and image in representation space via a pretext time prediction task, surpassing previous work in time-of-day estimation. The learned time-aware representations were further studied via validations on several downstream tasks. The strong performance in these downstream tasks highlighted its capability to recognise the similarity of the captured time (in time-based image retrieval), frame-coherent priors in TICL for video scene understanding (significantly improved video scene classification), and produce realistic and time-consistent performance in time-aware image editing (accurately reflecting typical lighting conditions for different times of day).

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

## Appendix Roadmap

This is the appendix for the main paper. Here is a general roadmap describing the contents of each part of this document supporting the main paper:

- We first provide additional details to the datasets in the Appendix A, which includes how we cleaned the originally noisy data into datasets that reflect the diversity of natural images paired with accurate metadata.

- In Appendix B, we cover the detailed illustration of the implementation of the model and the setup of the experiment. Along with additional performance and error analysis. We also provide additional results testing the ability to jointly predict date related metadata other than just the clock time.

- In Appendix C, we explore various scalar encoding methods to time variables on the pre-text task through an regression example in Appendix C.1. We also discussed the inherent trade-off of fine-grained classification via an additional ablation to the number of classes in Appendix C.2.

- Appendix D provides additional qualitative evaluation to the time-based image retrieval task.

- Appendix E gives experimental setup details, as well as more evidences of the intra-video consistency identified in the main paper in Appendix E.2.

- In Appendix F.1, we provide a detailed setup of the experiment along with additional qualitative and quantitative evaluation of the capability of time-aware features in image editing tasks.

- Appendix F.2 also shows results of using time-aware features to further improve the fidelity w.r.t. clock time via time-aware features under more advanced diffusion model baselines.

- In the main paper, we discussed about the implications of time-awareness in visual scene understanding, in Appendix G, we provide more examples of text query about the conceptual relations between clock time and scene/action/objects text embeddings.

## A    More Details on Datasets

### A.1    The proposed TOC dataset

**Comparison to previous works:**    In this work, we introduce a new benchmark dataset that combines images from the YFCC100M (Thomee et al., 2016) and Cross-View Time datasets (Salem et al., 2020). As we briefly summarised in Section 4, the major differences of our datasets to previous static image datasets featuring time-metadata lies in the view/appearance diversity and metadata correctness. Table 6 gives an overview of the differences. For previous generic social media image datasets, there have been persisting issues of unreliable timestamp metadata due to unsynchronized user/device activity. Apart from unreliable groundtruth, the visual appearances of time in some of the images are often undefined in non-photographic

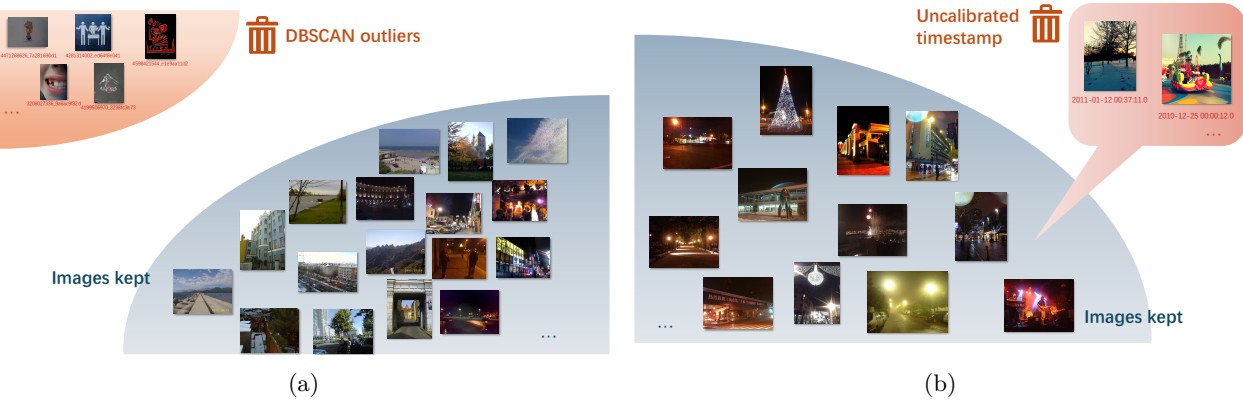

(a)                 (b)

Figure 10: **Dataset filtering process,** where (a) shows examples of finding unnatural images in DBSCAN (Ester et al., 1996) outliers that may degrade dataset quality, and (b) shows examples of removed images with uncalibrated clock timestamps.

Table 6: Comparison of existing image datasets with timestamps.

| Dataset | Image source | Timestamp reliability | Scene diversity |
|---|---|---|---|
| MIRFLICKR-1M (Huiskes & Lew, 2008) | mobile / miscellaneous | ✗ | high |
| YFCC100M (Thomee et al., 2016) | mobile / miscellaneous | ✗ | high |
| AMOS (Jacobs et al., 2007) | fixed webcams | ✗ | limited |
| CrossView Time (CVT) (Salem et al., 2020) | webcams + mobile devices | ✗ | mixed |
| **TOC (ours)** | **wild-view natural photography images** | **verified & timezone-aligned** | **high** |

images which do not reflect any natural time-of-day. Despite such these problems are mitigated for datasets with proportional samples from static surveilance camera, the repetitive views and occasional ground-truth leakage overlayed on the camera footage suggests limitity usablilty. These issues are depicted separately in Fig. 11. Therefore, our dataset resolved these issues by manually verifying metadata fidelity given the image on purely social media samples.

**Detailed curation steps:** Now, we cover more details of the dataset curation process. Fig. 10 gives a clear illustration of the data filtering steps to the dataset, improving the sample quality and metadata reliability. We firstly inspected all the night-time images with average pixel brightness $\geq 100$ to determine whether they have clearly mislabeled timestamps. Specifically, extreme cases like polar day were considered, so images with $|\text{altitudes}| \geq 75$ were retained regardless of illumination. This step removes clearly unsynchronized images, which do not align with human consensus about nighttime illuminations. To reduce the workload of filtering unnatural images, we firstly partitioned the images into 24 different hour intervals; within each of

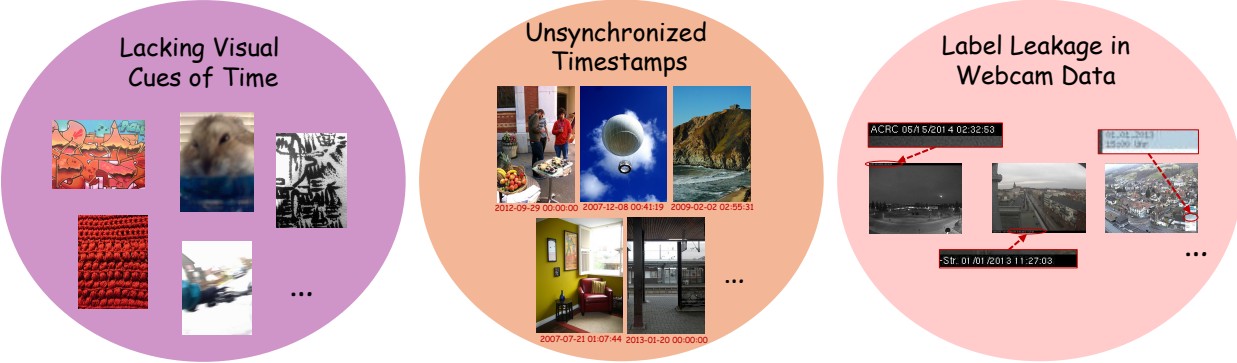

Figure 11: Existing issues in previous datasets (Salem et al., 2020; Thomee et al., 2016; Jacobs et al., 2007)

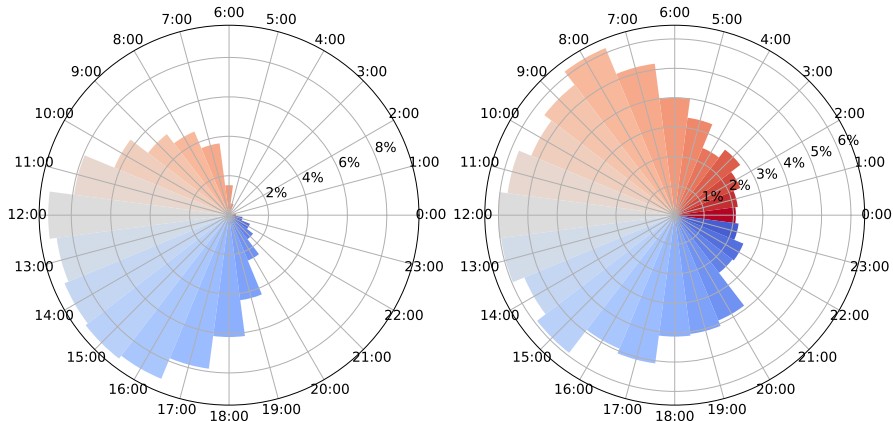

Figure 12: **Dataset hourly sample distribution,** where (a) shows hourly sample distribution for TOC dataset, in which daytime images are significantly more prevalent than nighttime images, and (b) shows hourly sample distribution for AMOS-test dataset displaying a similar skewed but more balanced distribution towards daylight hours.

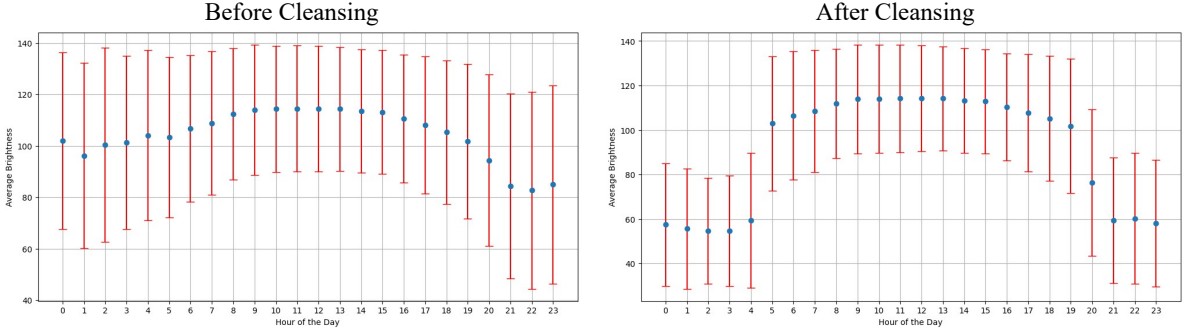

Figure 13: Mean brightness with standard deviation over images taken at different periods of the day.

them, we apply DBSCAN ($\epsilon = 10, \mathrm{minPts} = 100$) on ResNet-18 features, which gives a majority group and outliers. We recruited workers to manually review all the outlier images determining whether to add them back to the dataset. This pipeline allows for efficient removal of unnatural images without proper visual cues about time in the dataset by looking at samples groups with distinct features to the majority group containing natural photographs.

We conduct a statistical sanity check on the dataset–processing pipeline. The most intuitive visual cue correlated with the time of day is scene illumination, which we approximate by the statistical average of pixel brightness. In Fig. 13 we observe that, *before* cleaning, images labelled 00:00–06:00 in the previous dataset have brightness comparable to daytime photos, and an unnatural gap appears between the 21:00–23:59 and 00:00–03:59 bins. Both patterns violate common sense, where 1) late-night images should be markedly darker than daytime images, and 2) illumination should change most rapidly around sunrise and sunset, while remaining relatively stable during midnight and noon periods. These anomalies point to unsynchronised capture timestamps in the unprocessed dataset. *After* cleansing, the TOC dataset follows the expected smooth day–night trend, confirming that our pipeline yields data consistent with human intuition about illumination over a 24-hour cycle.

Following the data filtering, we partitioned the TOC dataset into a training set and a test set at a 9 : 1 ratio, with stratified sampling to ensure that the clock time distributions of both subsets were approximately equivalent. We observed a significant scarcity of images with reliable metadata captured at night compared

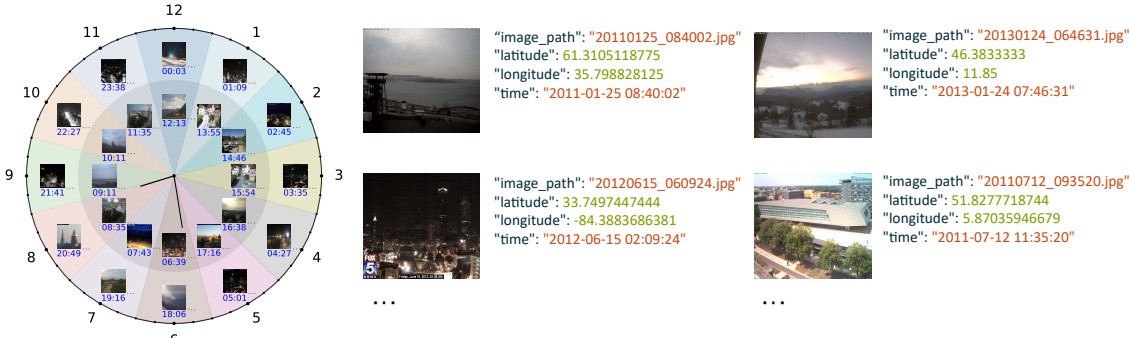

Figure 14: **Sample images from the AMOS test dataset.** The images showcase different scenes captured by stationary surveillance cameras at various times of the day with decent visual quality.

to daytime images. This observation corroborates our hypothesis that the distribution of timestamps in images shared on social media is inherently unbalanced as depicted in Fig. 12.

Such imbalance presents challenges in learning equitable embeddings for class time periods that are underrepresented due to limited sample availability. This imbalance necessitates strategic approaches to model training that can adequately compensate for these discrepancies.

## A.2 AMOS test dataset

**Dataset Filtering and SNR Estimation:** The AMOS-test dataset was selected from the CVT test set, containing 5,000 AMOS images, which was further reduced to 3,556 images. The dataset filtering involves several steps to ensure metadata reliability and sample quality. First, we calibrated the original UTC timestamps to their respective local timezones using the geolocation metadata. Then, we filtered out (1) noisy images with low Signal-to-Noise Ratio, where the SNR is estimated using a block-based variance method. Specifically, for an image $I$ with $N$ pixels, the SNR is computed as

$$\mathrm{SNR}(I) = 10 \cdot \log_{10}\left(\frac{\sigma_{\mathrm{signal}}^2}{\sigma_{\mathrm{noise}}^2}\right),$$

where the noise variance $\sigma_{\mathrm{noise}}^2$ is estimated as the average variance over the lowest 10% of non-overlapping blocks of size $16 \times 16$ pixels, and the signal variance is given by

$$\sigma_{\mathrm{signal}}^2 = \sigma_{\mathrm{total}}^2 - \sigma_{\mathrm{noise}}^2,$$

with $\sigma_{\mathrm{total}}^2$ being the variance of the entire image. Images with $\mathrm{SNR}(I) \leq 15$ were discarded. After cleansing, the average SNR improved from 1.93 (std = 10.35) to 3.38 (std = 3.50). This filtering ensured that only images with recognizable time-of-day related appearance were included in the evaluation. Figure 14 shows a few sample images from the dataset.

As the images were captured automatically by surveillance cameras with fixed views, the AMOS test set represents a different domain to the proposed TOC dataset. Although the dataset contains repetitive visual appearances due to the stationary setup of the cameras, it benefits from a more balanced distribution of timestamps throughout the day, as shown in Appendix A.1.

# B Implementation Details of TICL

In the main paper, we covered the high-level design of the TICL model we devised to learn time-awareness via a clock time estimation pre-text task. This section provides additional details.

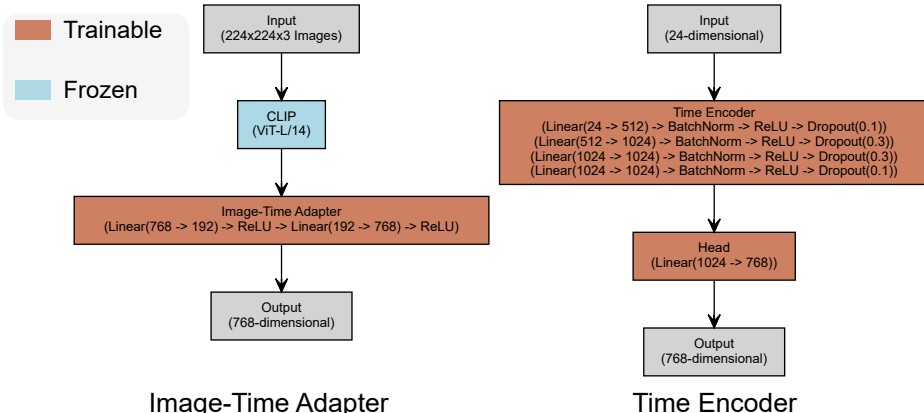

Figure 15: Visualisation of TICL sub-module architectures.

## B.1 Model details

**Time Encoder:** The Time Encoder consists of several fully-connected layers, with the detailed architecture shown in Fig. 15. The raw timestamps are first preprocessed into 24 one-hot class embeddings. The Time Encoder then takes these input class embeddings and projects them to the desired representation space.

**Image-Time Adaptor module:** The Image-Time Adaptor module is employed to adapt the raw backbone features with Time Encoder outputs, as depicted in Fig. 15. Training the Image-Time Adaptor module and Time Encoder jointly using a contrastive learning scheme allows for effective alignment between the two modalities.

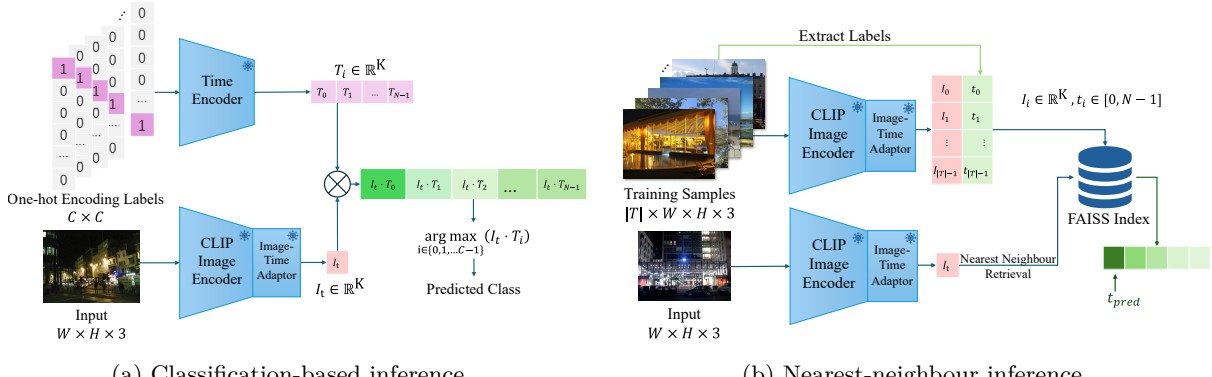

(a) Classification-based inference          (b) Nearest-neighbour inference

Figure 16: **Detailed illustration of different inference pipelines.** In (a), the model selects the clock time with the highest similarity to the input image from a finite set of clock time class embeddings. (b) shows that the model estimates clock timestamp by finding the corresponding timestamp of the nearest-neighbour to the input images from the training set based on the sample-specific TICL embeddings.

## B.2 Details in clock timestamp estimation inference pipelines

Two different clock timestamp estimation inference pipelines were devised. The first pipeline, shown in Fig. 16a, adheres to the classification scheme, selecting the timestamp with the highest similarity within a finite clock timestamp embedding pool encoded from $C$ one-hot embeddings. The second pipeline, shown in Fig. 16b, converts the problem to a retrieval-style formulation, using known image-timestamp pairs from the

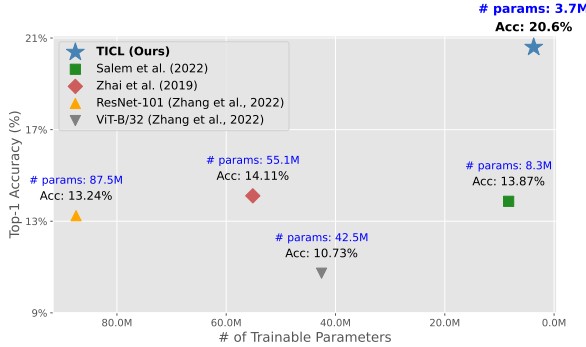

(a) Trainable parameters and performance comparisons.

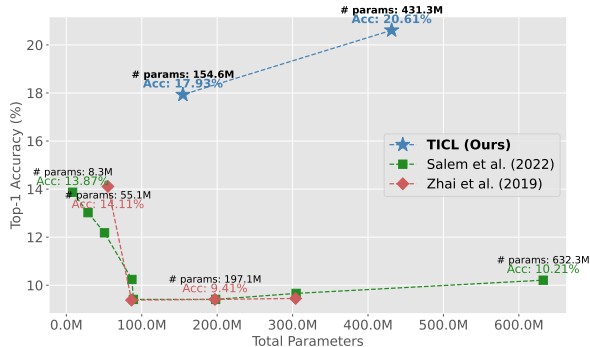

(b) Total parameters and performance comparisons.

Figure 17: **Parameter efficiency and performance.** (a) Comparison of trainable parameters and performance. (b) Analysis of total parameters and performance.

Table 7: **Joint time estimation performance on our TOC dataset.** Namely, we jointly estimate the month and hour using the same setup in the previous baseline Zhai et al. (2019) for fair comparison.

| | Hour Prediction | | | | Month Prediction | | | |
|---|---|---|---|---|---|---|---|---|
| | Top-1 acc ↑ | Top-3 acc ↑ | Top-5 acc ↑ | Time MAE (min.) ↓ | Top-1 acc | Top-3 acc | Top-5 acc | Month MAE |
| Salem et al. (2022) | 13.87% | 39.36% | 60.71% | 186.44 | 7.40% | 25.74% | 42.93% | 3.14 |
| Zhai et al. (2019) | 14.11% | 40.47% | 65.94% | 188.78 | 11.23% | 33.03% | 55.16% | 2.38 |
| Salem et al. (2022)[†] | 13.53% | 38.47% | 59.10% | 176.70 | 9.59% | 24.56% | 39.61% | 2.74 |
| Zhai et al. (2019)[†] | 15.01% | 42.54% | 68.24% | 185.34 | 12.03% | 35.91% | 60.50% | 2.25 |
| TICL (Hour only) | **20.60%** | **49.01%** | **67.82%** | 171.65 | - | - | - | - |
| TICL (Month only)[‡] | - | - | - | - | **34.48%** | **68.19%** | **82.88%** | **1.45** |
| TICL (Month, Hour) | 19.45% | 42.07% | 55.57% | 176.45 | 32.28% | 52.00% | 62.26% | 1.77 |

| | Hour Prediction | | | | Season Prediction | | | |
|---|---|---|---|---|---|---|---|---|
| | Top-1 acc ↑ | Top-3 acc ↑ | Top-5 acc ↑ | Time MAE (min.) ↓ | Top-1 acc | Top-3 acc | Top-5 acc | |
| TICL (Season, Hour) | 20.14% | 45.98% | 62.58% | **170.93** | 61.52% | 71.74% | 80.48% | - |

[†] These baselines take additional known geolocation metadata inputs, which boosted their performances on both prediction tasks.

[‡] Predicting 12 classes for months.

training set. The model returns the class-level timestamp of the most similar samples to it in the training set using an efficient vector search engine (Johnson et al., 2019).

**Settings for VQA baselines:** For baseline VQA based methods tested in Table 2, namely the BLIP (Li et al., 2022) and GPT-4o-mini (OpenAI et al., 2024), we simply use a pipeline with one round Q&A directly outputing times as predictions, which takes the input prompt $p_{\text{query}} =$ *"Estimate the LOCAL capture clock time of the image, answer with ONLY one 24-hour time in HH:MM format."*.

## B.3 Computational efficiency

Since the majority part of the TICL model, the CLIP image encoder, is frozen during training, the TICL training is thus efficient with a small number of trainable parameters. Figs. 17a and 17b shows that TICL achieved the best performance with the minimum trainable parameters among existing methods. Benefiting from the fewer trainable parameters, training on precomputed image features is significantly faster. Also, Fig. 17b demonstrates that simply scaling up the model parameters for previous works may even degrade the performance. We suspect that it is due to the more severe overfitting of the larger models on training samples. In comparison, the TICL model reached better performance with a moderate total number of parameters.

### B.4    Joint metadata estimation with time

We noticed that some of the previous baselines support joint time estimation instead of just focusing on clock time only. They often consider the joint contribution of other metadata including geolocation, date, and time of day to the image appearances (Salem et al., 2022; Zhai et al., 2019) to deal with the ambiguity of clock time when given only visual inputs. Therefore, in this section, we aim to explore such capability of estimating time and month jointly from only social media images.

We adjusted the network structure of TICL, enabling its ability to estimate month and hour jointly using a similar structure to Zhai et al. (2019), in which the model predicts $12 \times 24$ classes combining months and hours. We kept all the specified hyper-parameters and other setups the same. As for the compared baseline methods, we used the same hyper-parameters provided in previous works and picked the best performances from several trials, all the models are trained and tested on TOC dataset which contains only social media data to demonstrate the challenges on real-world samples. As provided in Table 7, TICL generally outperforms previous baselines when trained an tested on the more challenging TOC dataset without images with fixed views. In addition, under TICL paradigm, jointly predicting clock time and month or season does not provide boosts to individual tasks. We suspect that it is because of the gaps between the visual cues between the two different target variables. Such gaps lead to difficulties to model a joint probability of $P(t,m|x)$ for clock time $t$ and month $m$ with only the input $x$. However, the prediction advantages of models focusing on each attributes only suggest the possibility of stacking such different metadata-aware models in joint metadata verification-related tasks focusing on $P(t|m,x), P(m|t,x)$.

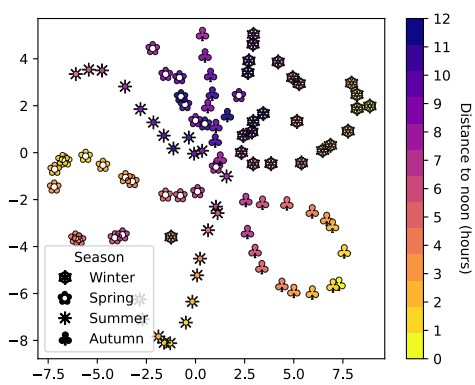

Figure 18: t-SNE visualisation of class embeddings `<clock time, season>`.

However, compared with using the date as additional supervision, the season supervision combines both date and geolocation gives a much smaller performance gap to the purely clock time supervised model. This suggests that learning meaningful combinations of other metadata could possibly provide a better clue to normalise the visual variance of the clock time. To support this point, we have included a visualization of learned clock time and season in terms of embedding in Fig. 18. As visualised, the model clearly recognizes season variations of the same clock time, while keeping a general pattern for different time periods of the day. This shows the potential of producing a possible calibration mechanism by learning on unnormalised timestamps with the help of different metadata supervision. This indicate a hopeful direction that we may learn an calibration mechanism to the raw clock time to bio-clock by incorporating comprehensive metadata and scene priors as supervision.

## C    Exploration of More Precise Time Encoding

### C.1    Scalar encoding

In this section, we explore limitations in a simple regression solution to the pre-text estimation task using scalar encoding of the clock time.

**Raw scalar encoding:**    The regression style construction for clock time estimation from images presents significant challenges as covered in main text. There are different issues with regression models, including 1) loss function sensitivity and 2) discontinuity in the scalar range for regression. In the following paragraphs, we first provide a brief illustration of the issue on the regression loss function. Secondly, we present experiments of a regression model working in a circular space instead of the vanilla scalar range which is a disconnected set (Zhou et al., 2019). These experiments provide explanations for the limits of vanilla regression models.

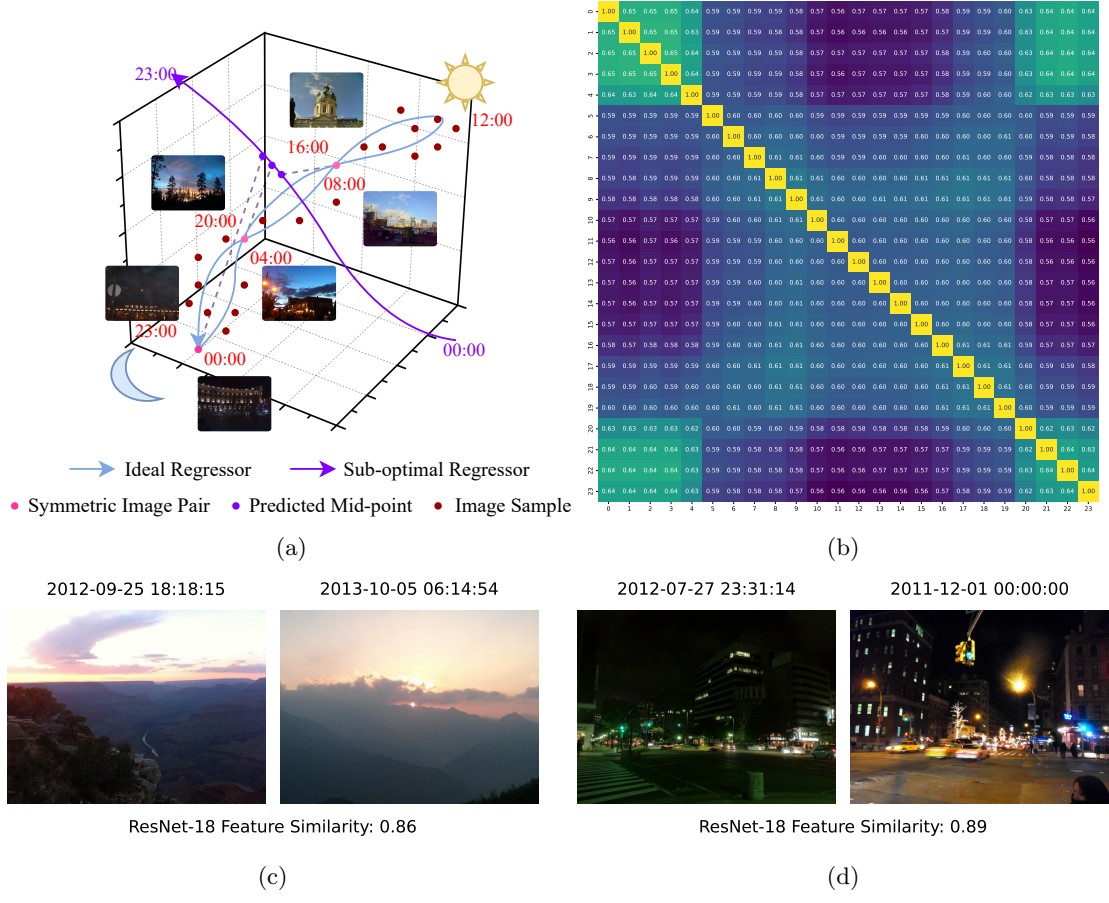

Figure 19: **Visual ambiguities for ground truth in regression.** (a) depicts a sub-optimal regression model where the predictions are biased towards the mid-point, and (b) shows a trend that images with more similar ResNet-18 features could have disparate timestamps. Few examples of such cases are provided in (c), (d).

Let us define the problem setting of clock timestamp regression as follows. Given an image $x$, the objective is to predict the timestamp $y$ in the range $[0, 24)$ hours of the day. In a regression framework, the model $f_\theta$ maps an input image $x$ to a continuous scalar output $\hat{y} = f_\theta(x) \in [0, 24)$.

Consider a dataset $\mathcal{D}$ consisting of images taken at various times throughout the day. Specifically, consider pairs of images $\{(x_i, y_i), (x_j, y_j)\}$ taken during "symmetric times" such as sunrise and sunset, where the general light conditions are similar but the ground truth timestamps are different (see Fig. 19d and 19c). With very similar inputs and the same model $f_\theta(\cdot)$, it holds that:

$$f_\theta(x_i) \approx f_\theta(x_j)$$

Then the Mean Squared Error (MSE) loss for the regression model over the dataset is defined as:

$$\mathcal{L}_{\text{MSE}}(\theta) = \frac{1}{|\mathcal{D}|} \sum_{k=0}^{|\mathcal{D}|-1} (y_k - f_\theta(x_k))^2$$

To find the optimal model parameters $\theta^*$, we minimise this loss function. Ideally, the goal of the optimiser is:

$$\nabla_\theta \mathcal{L}(\theta) = 0$$

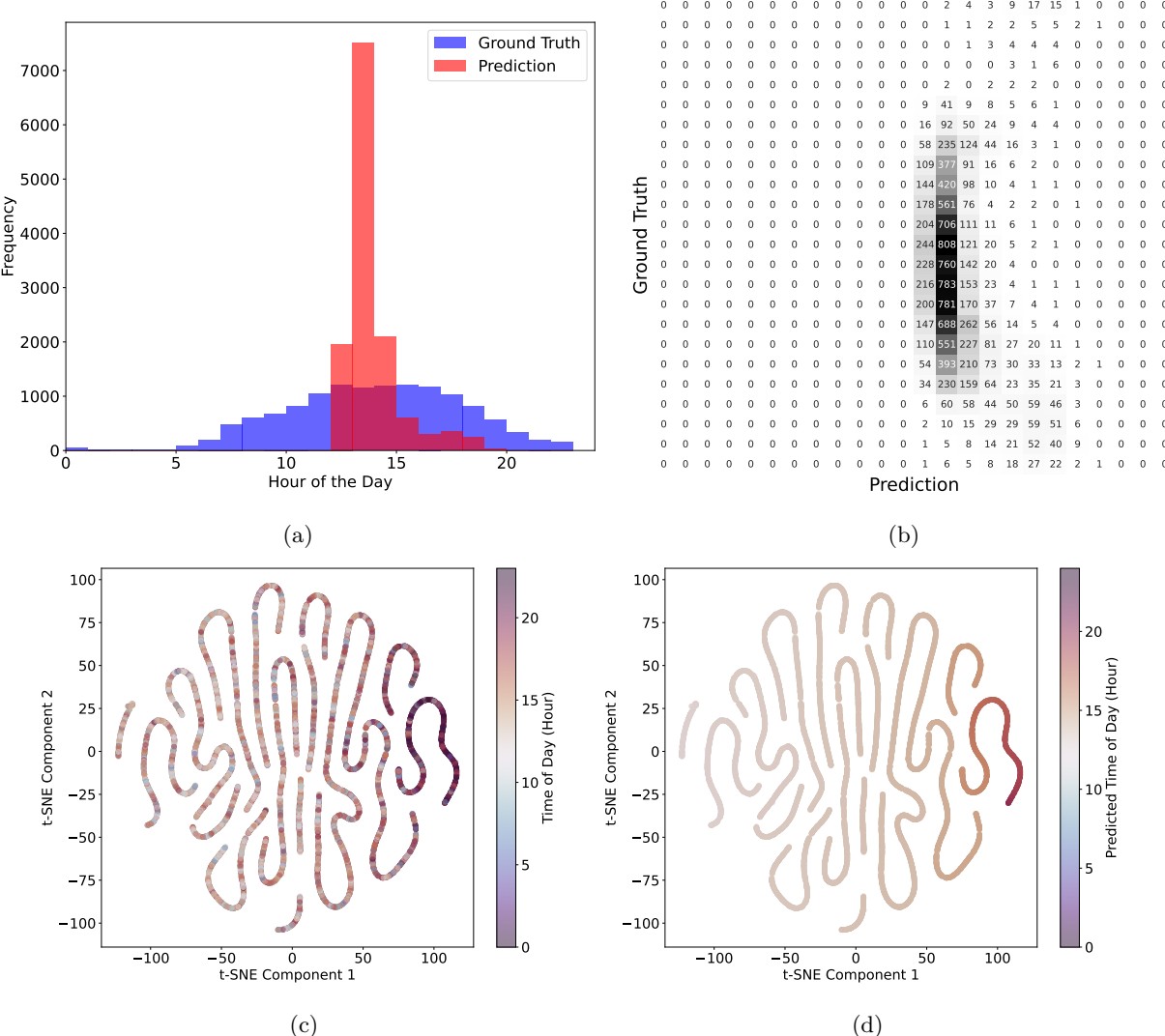

Figure 20: **Experiments on regression model.** (a) shows prediction distribution of regression model on TOC test set, (b) represents the confusion matrix by hour, (c) and (d) visualise t-SNE of regressor representations annotated with ground truth and predicted timestamps, respectively.

For pairs of similar images with different $y$, this optimisation leads to mid-point predictions:

$$\hat{y}_i \approx \hat{y}_j \approx \frac{y_i + y_j}{2}$$

This effect leads to local minima in the clock timestamp embedding space in Fig. 19a, particularly when $y_i$ and $y_j$ are at opposite ends of the 24-hour cycle, for example, 00:00 and 23:59. The regression model struggles with the ambiguous nature of time, resulting in systematically biased predictions towards the midpoint of symmetric clock times. Such bias results in incorrect gradient updates that cannot lead to an accurate estimation model for inputs $x_i, x_j$.

The aforementioned phenomenon of similar images with disparate ground truth timestamps prevails in the dataset. As evidence, we visualise the similarity of features using the ResNet-18 backbone throughout hours for the entire dataset in Fig. 19b. Therefore, this overall trend of feature similarity extends the reasoning to the entire dataset, where the predictions $\hat{y}$ are systematically biased towards the average of the whole clock time distribution. The predictions are likely to follow the normal distribution with the same mean value to

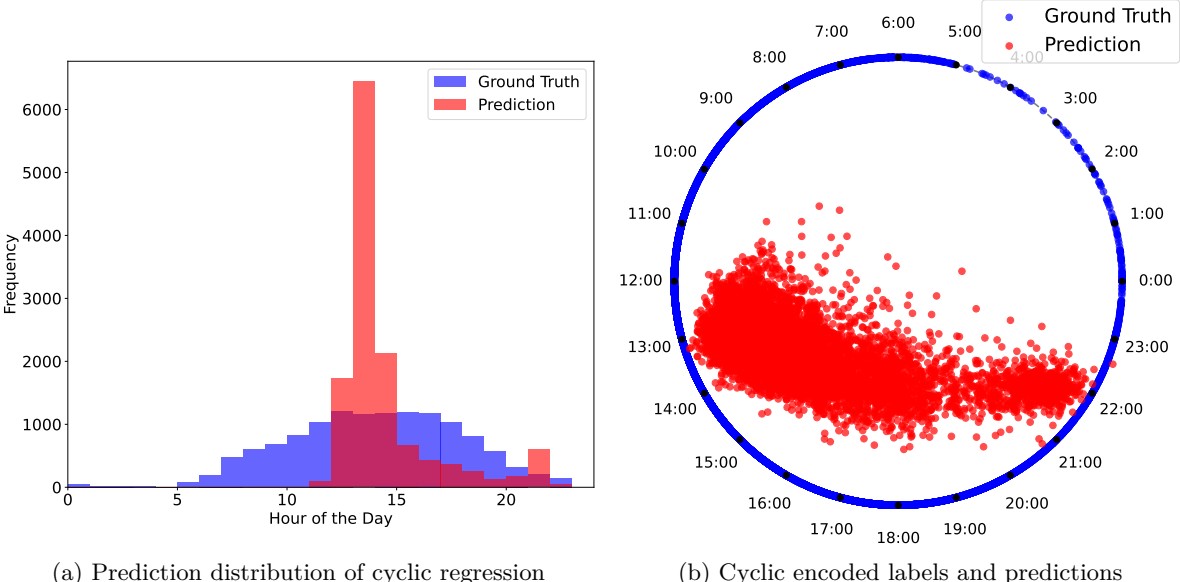

(a) Prediction distribution of cyclic regression    (b) Cyclic encoded labels and predictions

Figure 21: **Cyclic regression model results,** (a) shows prediction distribution of cyclic regression model, and (b) visualise how the cyclic encoding of predictions differ from the ground truth.

the ground truth distribution and smaller variance $\sigma$ (Murphy, 2012).

$$\hat{y} \sim \mathcal{N}\left(\frac{1}{|\mathcal{D}|}\sum_{k=0}^{|\mathcal{D}|-1} y_k, \sigma\right)$$

We conduct corresponding experiments to provide evidence for the claims above. Particularly, we train a regression model using ResNet-101 backbone. The prediction histogram and confusion matrix provided in Fig. 20a and Fig. 20b support our claims. The predictions are heavily concentrated around the average value of the ground truth distribution, while the actual timestamps in the dataset are more evenly distributed throughout the day. This discrepancy highlights the failure of the regression model to capture the cyclic nature of time, resulting in biased predictions of the average of the whole range. Fig. 20c shows that the regression model fails to discern similar images with different timestamps, where the features form disjoint trails on which images features from totally different time periods are nearly overlapped with each other. Fig. 20d further shows how the regression model predicts average timestamps for these images with similar features. These phenomena show that although the regression model managed to learn a certain extent of continuity of time of day from static views, it failed to tackle the ambiguity of clock timestamp given visual inputs with similar illuminations. Therefore, while such a regression model reaches convergence at local minima for the MSE loss, it is not ideal resorts we are looking for.

**Cyclic vector encoding:** As we identified in the main paper, the regression range for clock timestamp is a disconnected set. Here we present an attempt to solve the discontinuity of the clock timestamp scalar range: we adopted a previous method bridging the gap by trigonometric encoding and decoding to cyclic data (Adams & Vamplew, 1998). Specifically, it encodes the scalar data $y$ into points on the unit circle $(\cos{(y/y_{max})}, \sin{(y/y_{max})})$, and decodes the model outputs by reversing this process. Such representation space is proved to be continuous (Zhou et al., 2019). It bridged the gap between the end and the start of the regression value range, which was supposed to be close. We tried this remedy and found that it slightly mitigates the issue of over-concentration on the average values, as shown in Fig. 21a.

However, although this modification managed to rescue part of night images that are wrongly predicted toward the mean value of the whole target value range, it still exhibits poor prediction fairness, with most of the predictions falling in certain short time spans. The possible cause for such phenomena could still be the

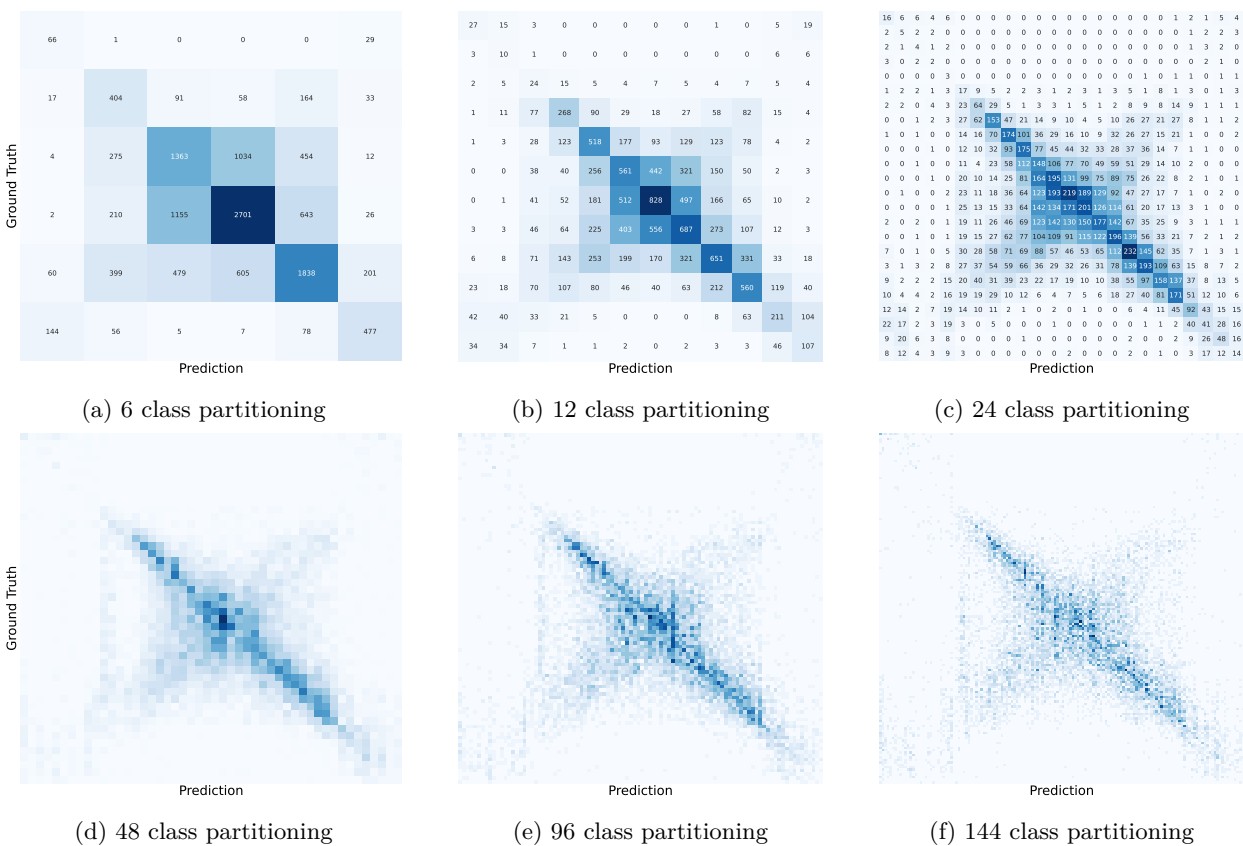

(a) 6 class partitioning      (b) 12 class partitioning      (c) 24 class partitioning

(d) 48 class partitioning      (e) 96 class partitioning      (f) 144 class partitioning

Figure 22: **Confusion matrices under different number of classes** provide more in-depth comparison of clock timestamp estimation performance.

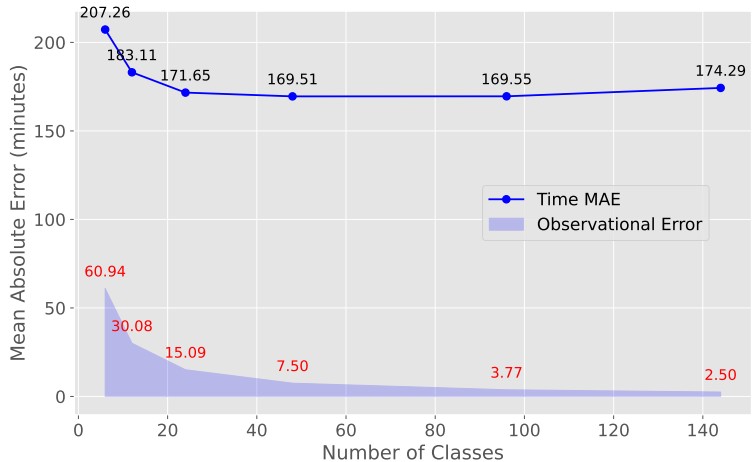

Figure 23: **Comparative error analysis of different class partitioning schemes,** it shows trends of mean absolute error (MAE) and observational error.

local minima that persist in the MSE loss landscape due to the prevailing timestamp ambiguities we discussed. Another observation in Fig. 21b is that there exists an obvious gap between the distribution of trigonometric encoding of ground truth timestamps and the predictions. This suggests that the cyclical correlation between

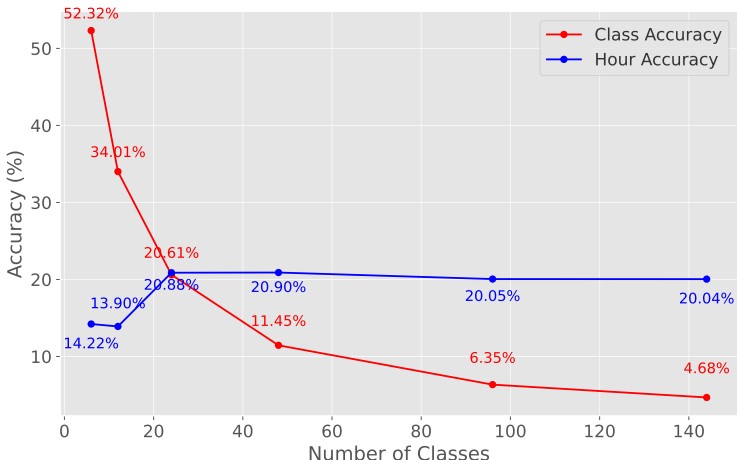

Figure 24: **Increasing number of classes does not further improve the time prediction accuracy.**, Class Accuracy represents the raw classification accuracy, and Hour Accuracy is calculated by $\frac{1}{|D|} \sum_{i=0}^{|D|-1} \mathbb{K}_{\left(\left\|\hat{Y}_i - Y_i\right\|_1 \leq 30 \text{ minutes}\right)}$, $\hat{Y}_i, Y_i$ are prediction and ground truth timestamps correspondingly.

visual appearances and clock time may not perfectly follow the simple unit-circle assumption in Adams & Vamplew (1998). In contrast, our proposed learnable embeddings for the target clock time labels in TICL can capture more complex correlations between different periods of clock times and visual cues without imposing such assumptions.

Therefore, the regression approaches struggle to properly address the ambiguity between time and visual features. Regression-based solutions are thus not as favourable for the pretext task of image clock time estimation.

## C.2 Ablation study on class partitioning

In the main paper, we adhere to the 24-class classification scheme used in previous methods. As loss of precision may introduce observation errors, we explore the effects of different granularities of class partitioning on pretext tasks.

To measure the precision loss, we compute observational errors, which are the average difference between actual timestamps and the converted class timestamps. Fig. 23 shows the mean absolute error (MAE) and the observational errors for different partitions of classes. As a part of MAE, observational errors are inherent such that they persist even with perfect class predictions (Conforti et al., 2020). Specifically, a small number of classes induces larger MAE, which is reasonable since converting actual timestamps to coarser time-span classes introduces larger additional observational errors.

However, this does not imply that extremely fine partitions should always be used to reduce observational error. We find that finer class partitioning, such as 144 classes, does not further improve the performance. In particular, Fig. 22 presents the performance of the TICL model on the TOC test set under different class partitioning. The overall distribution of predictions exhibits similar patterns despite different granularities. Fig. 24 highlights both class accuracy and hour accuracy for the model. The visualisation shows that while class accuracy drops significantly as the number of classes increases, the overall hour accuracy remains stable once the number of classes exceeds 24. This degradation in class accuracy with finer partitioning can be attributed to the smaller sample volumes within each class. The smaller the sample volume for each class, the more under-represented it tends to be (Sangalli et al., 2021). This suggests a potential drawback of finer class partitioning for downstream tasks involving time class embeddings.

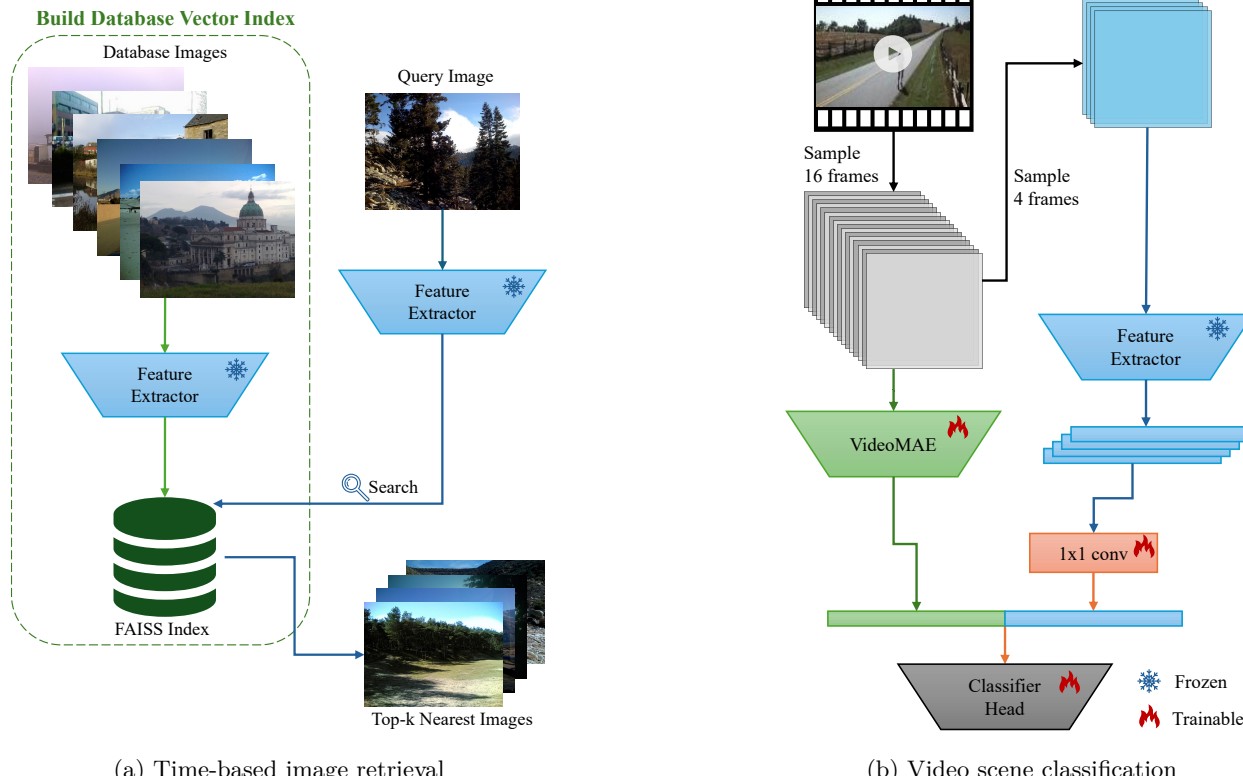

(a) Time-based image retrieval                    (b) Video scene classification

Figure 25: **Zero-shot downstream pipelines.** (a) corresponds with experiment pipelines for retrieval in the main paper, which is a zero-shot vector search engine for same-hour images based on FAISS (Johnson et al., 2019), and (b) shows one of the pipeline for video scene classification in addition to linear probing, in which we test the capabilities of TICL by plugging in the corresponding models to the feature extractor whose outputs are convoluted and concatenated to the backbone features (Tong et al., 2022).

Since the difference between clock time estimation performance of the 24-class partition and the optimal result achieved with different class partitioning is within an acceptable range, we choose the 24-class partition as the default in our main work. This choice allows for a fair comparison against previous methods, to ensure that our improvements are due to the proposed techniques rather than variations in class partitioning. Additionally, the 24-class partitioning, which reached Class Accuracy ≈ Hour Accuracy, also ensures that each class can be assigned enough samples so that a robust time class embedding could be learned.

To sum up, the ablation study on number of classes indicates that while the proposed TICL method can easily be extended to finer class partitioning schemes and maintains good hour accuracy and MAE, moderate granularity in class partitioning yields the best results for time estimation tasks. This supports our choice of a 24-class partitioning scheme for consistent benchmarking to previous baselines and verification of our conjecture on visual time awareness.

### C.3    Exploration on other contrastive learning methods

Given the ambiguity of visual appearances with respect to time of day, in addition to the vanilla InfoNCE contrastive learning we applied, another relaxed supervised contrastive learning counterpart is also tested to learn the TICL model following Segsort (Hwang et al., 2019). Concretely, we replace the InfoNCE objective with an hour-aware, cluster-based pulling/pushing scheme: for each hour, embeddings are repeatedly clustered with $k$-means; samples that fall into the *same* hour-specific subcluster within a minibatch (memory bank)

are pulled together, while others are pushed apart via a negative log-likelihood objective (as in Hwang et al. (2019)). At inference, we embed a query image once and predict clock time via cosine $k$-nearest-neighbour over the frozen gallery of training embeddings. In the experiment below, We follow defaults in Hwang et al. (2019): the hour-wise subcluster count is set to $n_{\mathrm{sub}} = 25$, the inner $k$-means refinement uses `max_iter`$= 10$ iterations per refresh, and the assignment sharpness is controlled by `concentration_constant`$= 10$. All remaining training and optimization settings are identical to those in TICL.

Table 8: TOC and AMOS test results under different contrastive learning methods.

| | TOC | | | | AMOS | | | |
|---|---|---|---|---|---|---|---|---|
| **Contrastive Loss** | Top-1 acc ↑ | Top-3 acc ↑ | Top-5 acc ↑ | Time MAE (min.) ↓ | Top-1 acc | Top-3 acc | Top-5 acc | Time MAE (min.) |
| Segsort-style (Hwang et al., 2019) | 18.97 | 39.32 | 52.27 | 171.55 | 12.29 | 28.37 | 39.57 | 209.00 |
| InfoNCE (Classification) | 20.60 | 49.01 | **67.82** | 171.65 | **13.55** | **38.49** | **57.28** | **187.87** |
| InfoNCE ($k$NN) | **25.67** | **49.32** | 66.74 | **156.24** | 11.14 | 31.01 | 48.84 | 220.94 |

As shown in Table 8, across both datasets, the Segsort-style variant did not consistently improve over InfoNCE. On both TOC AMOS, InfoNCE methods achieves the best performance. We hypothesize two contributing factors: (i) prototype assignments in Segsort are well-suited to separable semantic classes, whereas time-of-day differences can be subtler than object-level semantics; (ii) hour-wise $k$-means over CLIP features tends to form subclusters by object identity early in training, which may misalign with temporal structure and provide weaker supervisory signals. Developing supervised contrastive objectives that more directly encode temporal neighbourhoods remains a promising direction; we therefore retain InfoNCE as the default in TICL.

## D    Qualitative Time-based Image Retrieval Results

Fig. 26 provides a closer look at the retrieved images using the pipeline in Fig. 25a as part of a more detailed qualitative evaluation of retrieval performance. Some of the retrieved images have totally different content from the query images, but share similar light conditions. This suggests that our model disentangles the time-awareness from rich semantics of CLIP representations, which have more semantic focus to the subjects. In addition, the negative predictions still share similar illumination to the query images, suggesting the essence and ambiguity of clock time to visual appearances.

## E    Additional Results on Video Scene Classification

### E.1    Experiment setup

The performance of different models on the video scene classification task was evaluated across three datasets, each containing videos with distinct styles. Apart from simple linear probing, we also tested the model's performance fused with/against a baseline method VideoMAE (Tong et al., 2022). The detailed fusion architecture is visualized in Fig. 25b

- **Hollywood2-Scene** (Marszałek et al., 2009) is a movie clip-based dataset with 570 training videos and 582 test videos across 10 scene classes, totalling 20.1 hours. Each video represents a specific dramatic scene with multiple shots, meaning drastic view/subject changes within.

- **YUP++** (Derpanis et al., 2012) comprises 1200 videos across 20 scenes captured by either stationary or moving cameras. Given the significant differences between the 20 scenes and the fact that the average clip duration is only 5 seconds, the classification task on it is considered less challenging (Wang & Koniusz, 2023).

- **360+x** dataset (Chen et al., 2024) is a more recent dataset introduced for holistic dynamic scene understanding with multiple views captured by stationary cameras. It consists of 15 indoor scenes and 13 outdoor scenes, with 1380 clips totalling 67.78 hours. Its multi-view and stationary camera traits enable us to evaluate how our learned time-awareness perform on different types of views individually.

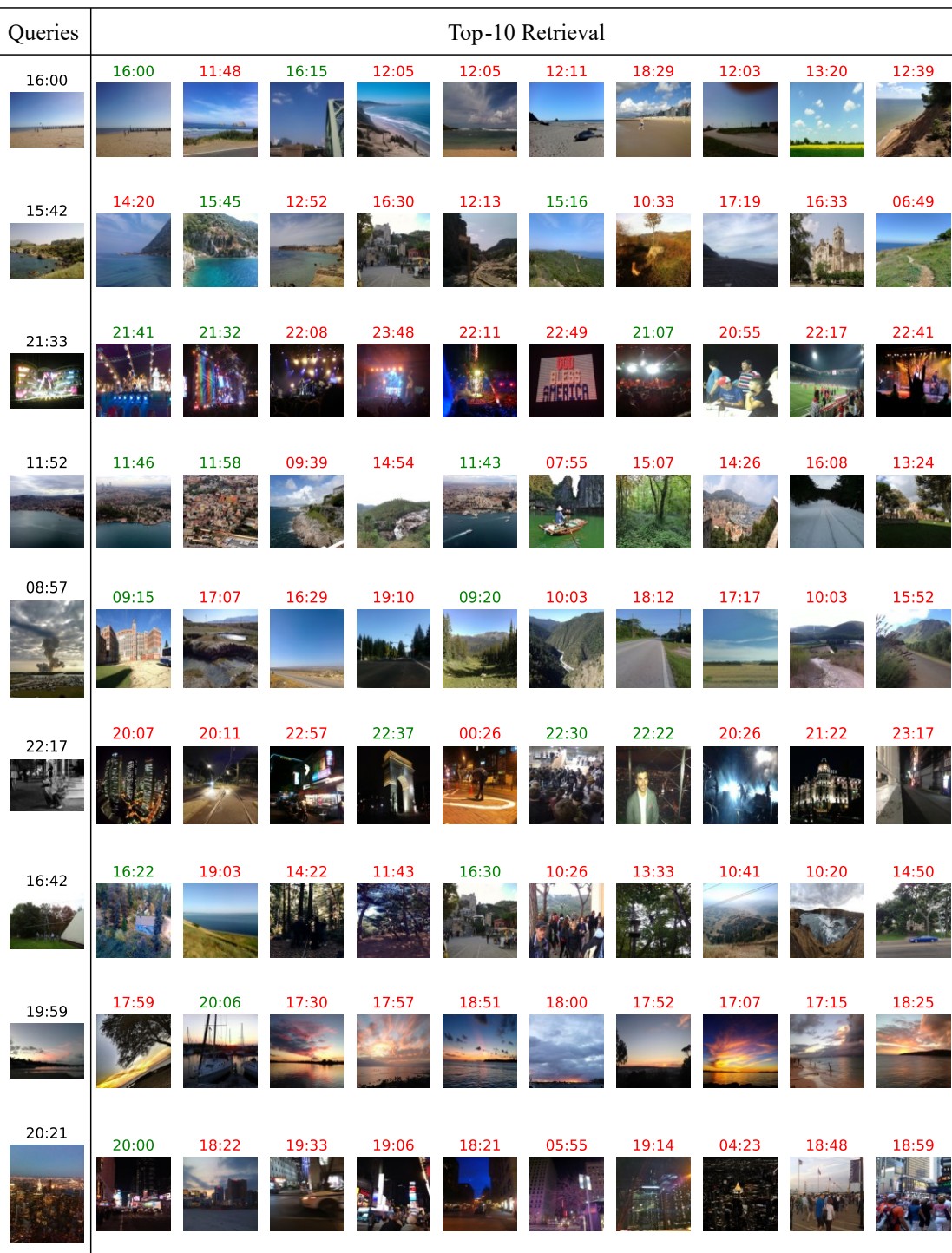

Figure 26: **Randomly sampled retrieval results.** Each image is annotated with its corresponding timestamp, green captioned images are positive retrieval while red are negative predictions with Error > 00:30, retrieved images closer to the left have larger similarity to the query images.

**Hyper-parameters:** For fair comparison, a fixed set of hyper-parameters was used in different experiment trials. Apart from the number of epochs and the learning rate, we followed all the parameter settings in Tong et al. (2022). And we only varied numbers epochs and learning rate for different datasets in. We report

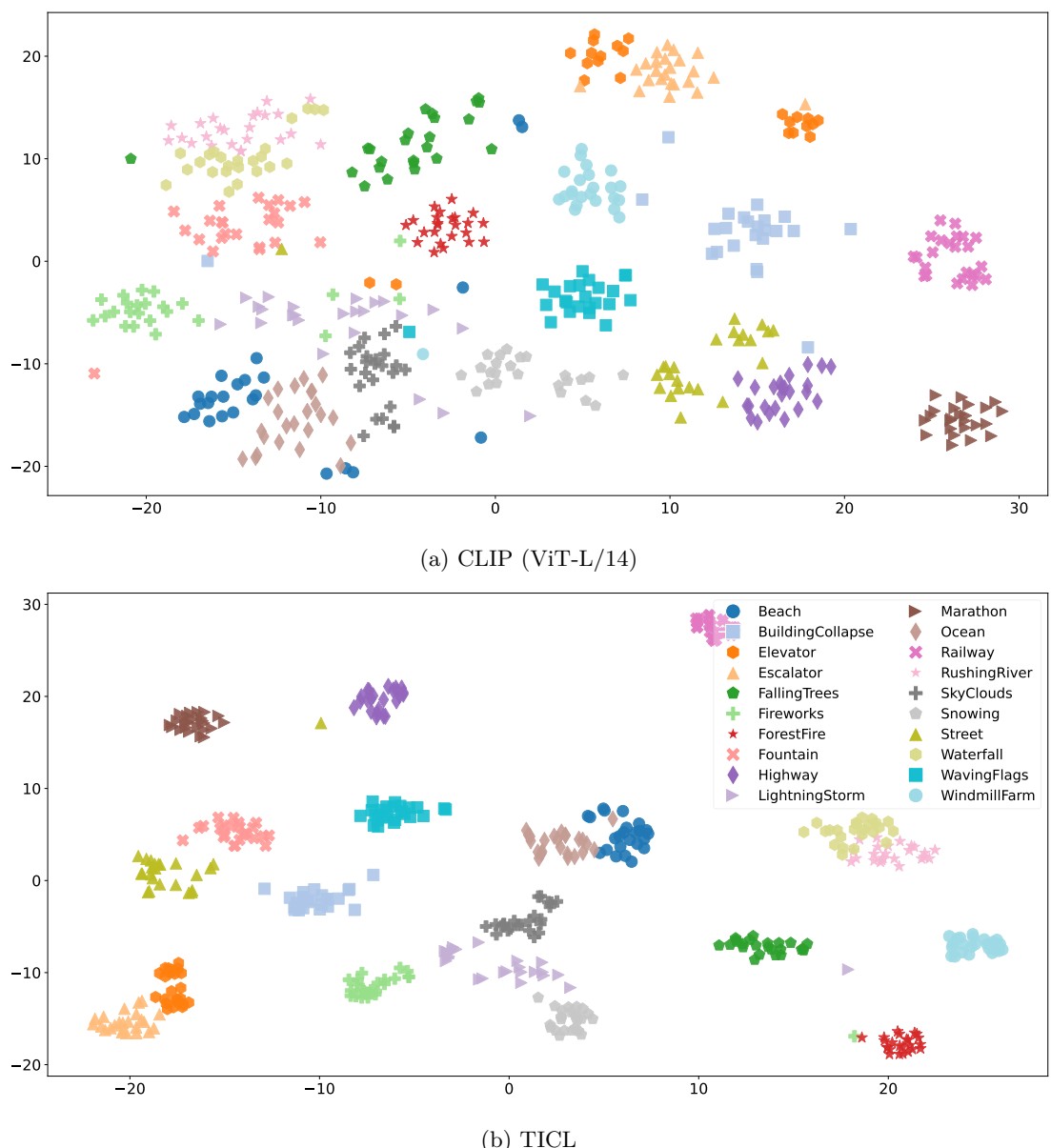

(a) CLIP (ViT-L/14)

(b) TICL

Figure 27: **t-SNE visualisation comparison.** It visualises time-aware video features in YUP++ dataset (Derpanis et al., 2012). Each embedding is annotated by their corresponding labels. It exhibits a similar trend to the t-SNE results in the main paper.

the best result achieved for each method tested. Specifically, a training/validation split of 5:1 was applied to each original training dataset to fairly select the best checkpoints for each method.

### E.2 Time embedding coherence on video frames

As discussed in the main paper, the observed improvements when integrating time-aware features with video classification backbone models could be attributed to the stronger intra-video consistency of these time-aware features.

To provide quantitative evidence of this consistency, we examine the characteristics of time-aware features across frames within each video. The backbone VideoMAE (ViT-B) model takes the input by sampling 16

Table 9: Hyper-parameters used for video scene classification on different datasets.

| Hyper-parameter | Hollywood2-Scene | YUP++ | 360x (Third-person) | 360x (Panoramic) |
|---|---|---|---|---|
| Learning Rate | $5 \times 10^{-5}$ | $5 \times 10^{-5}$ | $7 \times 10^{-5}$ | $7 \times 10^{-5}$ |
| # Iterations | 20 | 10 | 20 | 20 |
| **Default Settings from (Tong et al., 2022; Wolf et al., 2020) (Common Across All Datasets)** | | | | |
| Optimizer Type | adamw_torch($\beta_1 = 0.9, \beta_2 = 0.999, \epsilon = 10^{-8}$) | | | |
| LR Scheduler | linear | | | |
| Batch Size | 2 | | | |

Table 10: **Mean intra-video feature variance.** It is computed by the mean feature variance of 16 input frames for each video using different models, showing a quantitative evidence of intra-video feature consistency of time-aware models.

| Models | Hollywood2-Scene | YUP++ | 360+x (Third-person) | 360+x (Panoramic) |
|---|---|---|---|---|
| CLIP (ViT-L/14) | $7.49 \times 10^{-2}$ | $2.49 \times 10^{-2}$ | $3.31 \times 10^{-2}$ | $2.83 \times 10^{-2}$ |
| Salem et al. (2022) | $3.52 \times 10^{-6}$ | $1.23 \times 10^{-6}$ | $7.55 \times 10^{-7}$ | $7.86 \times 10^{-7}$ |
| Zhai et al. (2019) | $2.50 \times 10^{-4}$ | $1.00 \times 10^{-4}$ | $8.50 \times 10^{-5}$ | $7.59 \times 10^{-5}$ |
| TICL (Ours) | $3.33 \times 10^{-4}$ | $1.24 \times 10^{-4}$ | $1.44 \times 10^{-4}$ | $1.33 \times 10^{-4}$ |

frames evenly from each video. For the 16 input frames, we observed that the time-aware features of these 16 frames exhibit significantly smaller average variance compared to their CLIP features, as shown in Table 10.

This finding supports our intuition that a natural video that depicts a dynamic scene is typically captured over a short period of the day, leading to relatively small changes in the time-aware features of consecutive frames. In contrast, the CLIP features show more drastic changes between frames, making it harder to summarise consistent frame-wise features into coherent video-level features. The t-SNE visualisation comparisons to these features in main paper and Fig. 27 provide additional results to prove that TICL video features are more separable than CLIP video features.

Thus, time-aware feature extractors provide more consistency across different frames, making it easier to capture time-related visual priors in videos, which correlate with scene categories. These time-aware video priors eventually improved the video scene recognition performance, as illustrated in the main text.

However, it is observed that the embeddings in Salem et al. (2022) and Zhai et al. (2019) have much smaller intra-video feature variances, but they perform worse than the TICL features we proposed. Given that the previous methods produce 128-dimensional time-aware embeddings, which dimensionality is much lower than TICL embeddings, it is expected that they have much smaller variances. Moreover, although previous methods perform moderately better than the baseline methods in the majority of test datasets, their performance degradation in panoramic video datasets suggests a limitation in terms of generalisation ability between different styles of videos, especially for those captured in rare camera views in the 360+x dataset (Chen et al., 2024). In contrast, TICL utilising a strong foundation model generalised better across different kinds of videos.

In summary, time-aware embeddings could provide a more coherent representation among multiple sequential frames in a video, which are relatively invariant to sudden view/object changes altering the semantic meaning of the frame. Among the time-aware models, TICL gives more robust time-aware priors that generally bring more improvements than all the other time-aware models on different styles of video.

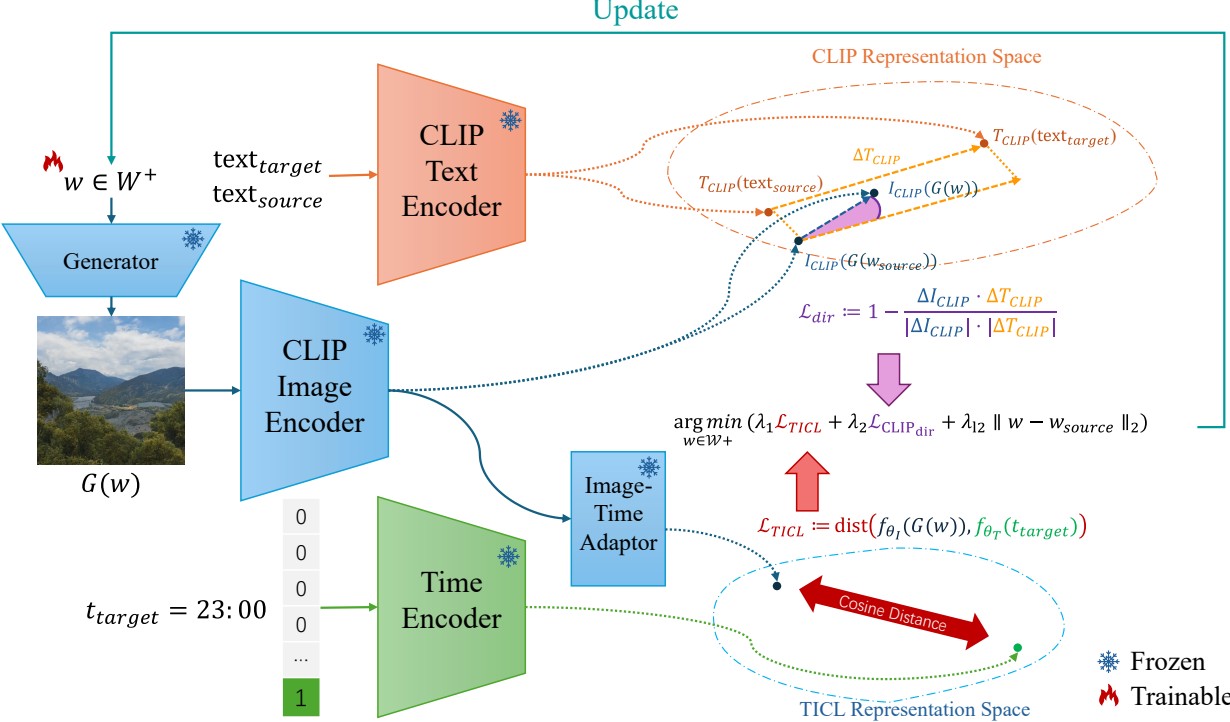

Figure 28: **Time-aware image editing pipeline.** This is the pipeline for latent optimisation for image editing, where $w, w_{source}$ represents latent vectors for ongoing edit outcomes and original images, $t_{\text{target}}$ is the one-hot encoding of the desired time of day for the output image, $G(\cdot)$ is the generator, $\text{dist}(\cdot, \cdot)$ computes the cosine distance between two vectors, $\Delta I_{\text{CLIP}}$ is the difference between CLIP embeddings of the original image, $\Delta T_{\text{CLIP}}$ stands for the difference between the source and target caption embeddings. $f_{\theta_I}(\cdot), f_{\theta_T}(\cdot)$ corresponds to components of the TICL model.

# F   Additional Results on Time-aware Image Editing

## F.1   Latent optimisation

**Experiment setup & Hyper-parameters:**   Fig. 28 gives an overview of the experiment pipeline we used for the time-aware image editing task. For the main paper's results, each columns results were obtained via the same hyper-parameter setup. Specifically, we set the target timestamps $t_{target}$ as visualized in the figure and fixed the $\lambda_1 = \lambda_2 = 1$, using `Adam` optimiser with `lr_rampup`= 0.05 for all experiments; we varied other hyper-parameters as visualized in the following Table 11 w.r.t. different target time periods of the day and the subject contents of images.

Table 11: Hyper-parameters for LHQ (Pinkney, 2024), LSUN-Church, and LSUN-Bedroom (Yu et al., 2015) editing processes.

| Hyper-parameter | LHQ | | | LSUN-Church | | | LSUN-Bedroom | | |
|:---:|:---:|:---:|:---:|:---:|:---:|:---:|:---:|:---:|:---:|
| | **Noon** | **Evening** | **Night** | **Noon** | **Evening** | **Night** | **Noon** | **Evening** | **Night** |
| $\lambda_{L2}$ | 0.001 | 0.001 | 0.0005 | 0.001 | 0.001 | 0.0005 | 0.001 | 0.001 | 0.0005 |
| # iterations | 50 | 50 | 100 | 50 | 50 | 100 | 50 | 50 | 100 |
| lr | 0.07 | 0.07 | 0.1 | 0.5 | 0.5 | 0.5 | 0.05 | 0.05 | 0.05 |

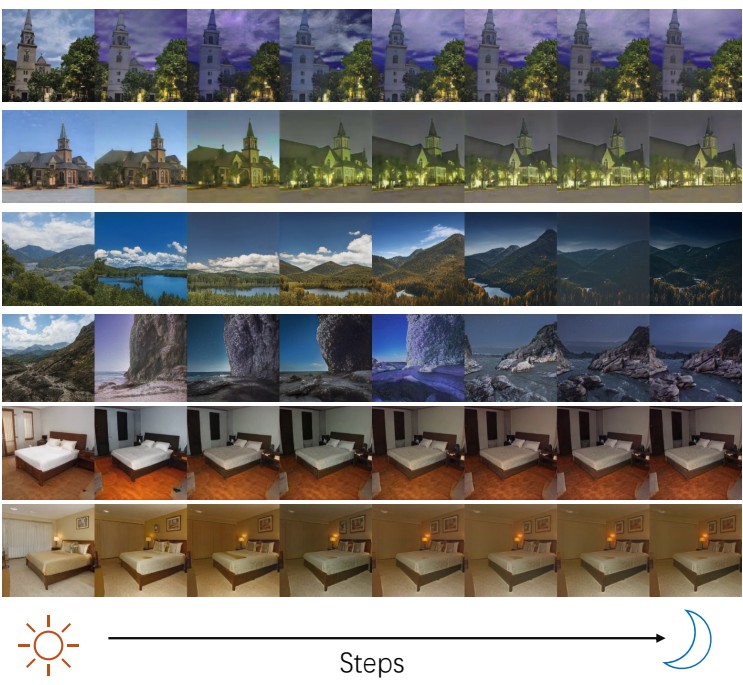

Figure 29: **Day-to-Night Edits.** An example of transitioning images from daytime to nighttime using latent optimization. This figure shows the progression of edits from various starting points to target times of day 22:00 (The rightmost figures are outputs for each edits.).

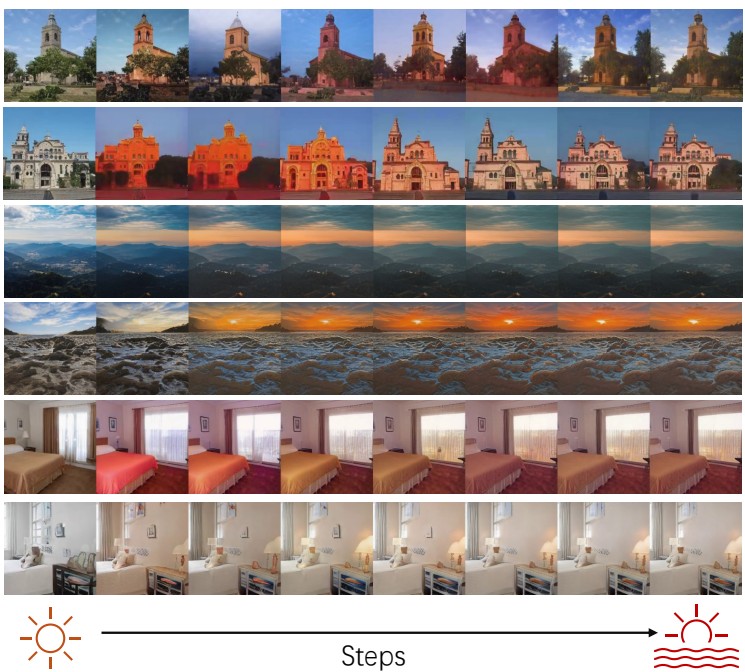

Figure 30: **Day-to-Evening Edits.** An example of transitioning images from daytime to evening using latent optimization. This figure shows the progression of edits from various starting points to target times of day 19:00 (The rightmost figures are outputs for each edits.).

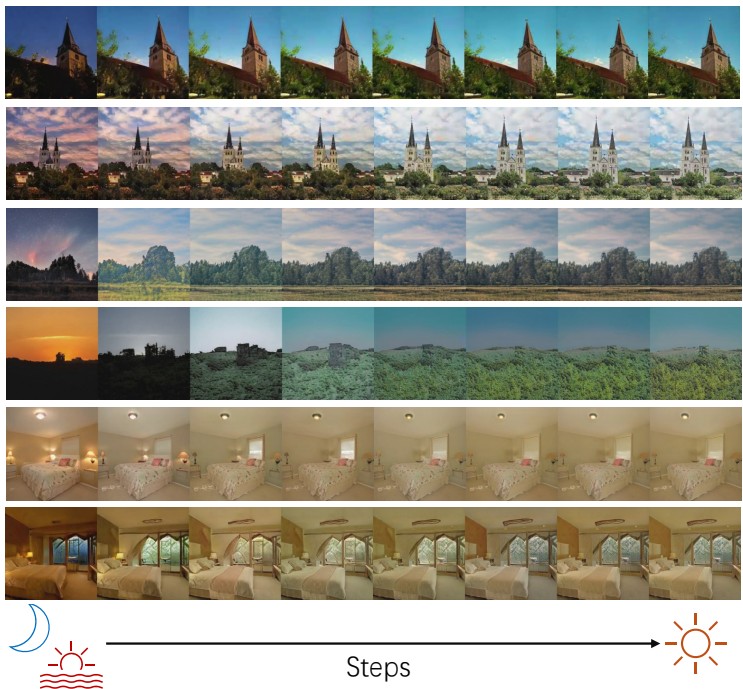

Figure 31: **Night/Sunset-to-Noon Edits.** An example of transitioning images from nighttime/sunset to noon using latent optimization. This figure shows the progression of edits from various starting points to target times of day 12:00 (The rightmost figures are outputs for each edits.).

Table 12: **FID Scores.** They quantitatively show how realistic the image editing results are for different methods on two image edit directions.

| Methods | Day-to-Night ↓ | Day-to-Sunset ↓ |
|---|---|---|
| Latent optimisation ($\mathcal{L}_{CLIP}$) (Patashnik et al., 2021) | 53.55 | 50.60 |
| Latent optimisation ($\mathcal{L}_{\text{CLIP}_{dir}}$) | 50.07 | 50.59 |
| **Latent optimisation ($\mathcal{L}_{\textbf{CLIP}_{dir}} + \mathcal{L}_{TICL}$)** | **48.97** | **50.41** |
| StyleGAN NADA (Gal et al., 2021) | 78.80 | 66.58 |
| CLIPStyler (Kwon & Ye, 2022) | 71.12 | 73.59 |

**More qualitative results:** Additional results of latent optimisation based editing are presented. We varied the initial latent vectors and target hours to show the broader capabilities of our approach. Fig. 29, Fig. 30 and Fig. 31 provide more examples of time-aware image editing with intermediate results during optimisation steps. The results suggest that our method could be applied to broad time-aware editing directions, which can start from images from various times of day.

**Quantitative evaluations (User study):** In addition to the qualitative evaluation results, we also include quantitative metrics to evaluate the synthesis results. Table 12 gives FID scores (Heusel et al., 2017) to different edit directions calculated by the official PyTorch implementation of Seitzer (2020) on 5000 samples for each methods. Our method outperforms existing methods with a smaller FID score suggesting more realism in the synthesised images. Additionally, we conducted a user study (by using the mean-opinion-score scheme) on the output images. The preference scores for each method are reported in Table 13, further demonstrating the advantages brought by incorporating time-aware embeddings.

Table 13: **User study evaluating image editing qualities,** in which we report preference scores and their standard deviation (in brackets). Preference scores range from 1-5, and higher scores mean better preferences.

| Methods | Day-to-Night ↑ | Day-to-Sunset ↑ |
|---|---|---|
| Latent optimisation ($\mathcal{L}_{CLIP}$) (Patashnik et al., 2021) | 2.80 (0.60) | 2.84 (0.53) |
| Latent optimisation ($\mathcal{L}_{\mathbf{CLIP}_{dir}}$) | 2.63 (0.85) | 3.28 (0.67) |
| **Latent optimisation ($\mathcal{L}_{\mathbf{CLIP}_{dir}} + \mathcal{L}_{TICL}$) (Ours)** | **3.34 (0.64)** | **4.01 (0.58)** |
| StyleGAN NADA (Gal et al., 2021) | 2.41 (0.89) | 2.36 (1.17) |
| CLIPStyler (Kwon & Ye, 2022) | 2.08 (0.62) | 1.81 (0.93) |

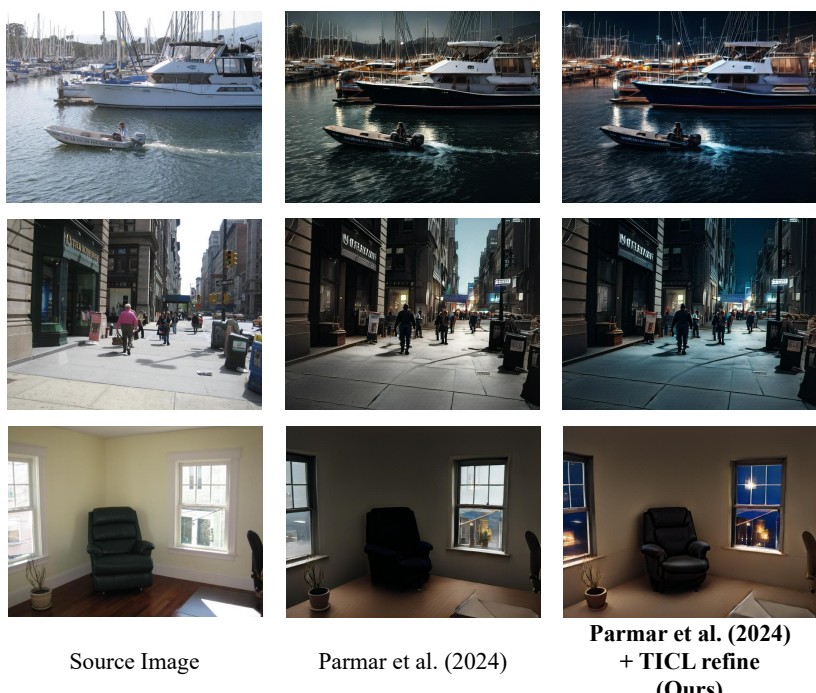

| Source Image | Parmar et al. (2024) | **Parmar et al. (2024) + TICL refine (Ours)** |

Figure 32: **Visualisation of Day-to-night edits (Part 1).** Transitioning images from day to night using the diffusion model.

## F.2  Editing with diffusion models

Given that the previous baseline latent optimisation image editing method has limited capabilities, we extend our experiment to a more recent editing method Parmar et al. (2024) using diffusion models (Ho et al., 2020; Rombach et al., 2021).

**Experiment setup & Hyper-parameters:**  Specifically, we optimise the edit target text embedding $E^*_{text}$ to minimise the cosine distance between the time-aware embeddings of the output images and the target clock timestamp embeddings, which is written as:

$$E^*_{text} = \arg \min_{E_{text}} \mathrm{dist}\left(f_{\theta_I}\left(G\left(x, E_{text}\right)\right), f_{\theta_T}\left(t_{target}\right)\right)$$

where $E^*_{text}$ is the target text embeddings for the text-based image editing model $G(\cdot, \cdot)$ takes input image $x$ and guidance text embedding $E_{text}$. $f_{\theta_I}, f_{\theta_T}$ corresponds to TICL model components. $dist(\cdot, \cdot)$ measures the cosine distance of two embeddings. It essentially optimises the guidance text embeddings $E_{text}$ to achieve better editing results that visually align with the target time.

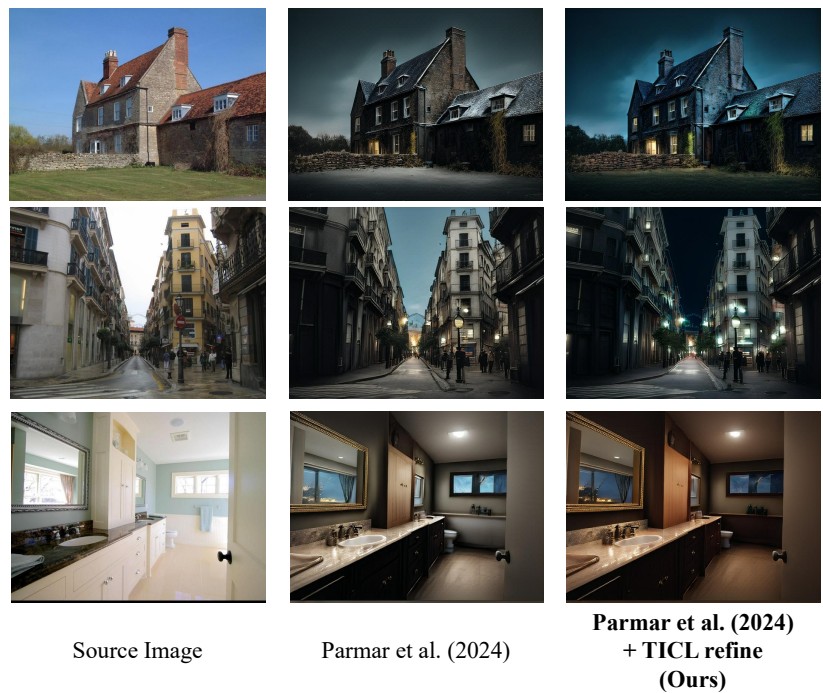

Source Image    Parmar et al. (2024)    **Parmar et al. (2024) + TICL refine (Ours)**

Figure 33: **Visualisation of Day-to-night edits (Part 2).** Continuing results of time-aware editing using the diffusion model.

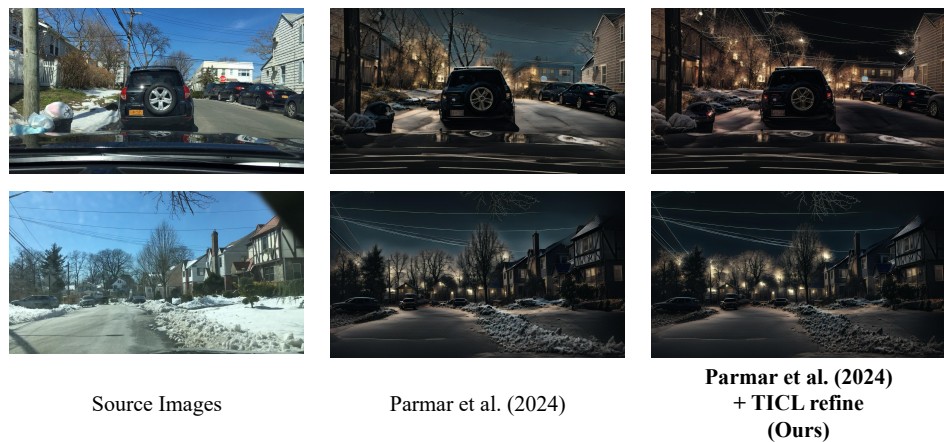

Source Images    Parmar et al. (2024)    **Parmar et al. (2024) + TICL refine (Ours)**

Figure 34: **Visualisation of Day-to-night edits (Part 3).** Further results demonstrating day-to-night transitions using the diffusion model.

As for hyper-parameters, we applied default experiment settings for the baseline editing process as provided in Parmar et al. (2024) with text guidance set to "`a photo of {target time period }`". The subsequent optimisation process to $E_{text}$ uses `Adam` optimiser with learning rate = 0.02 and 10 iterations without any further configuration.

**Qualitative results:** As shown in Fig. 32, Fig. 33 and Fig. 34, although additional optimisation steps for each edit are required, it refines the existing method with more reasonable synthesis results compared with using purely text editing guidance, further proving the general applicability of the TICL embeddings to the whole image-editing subfield.

# G   Additional Text Queries on Clock Time Class Embeddings

In video scene classification tasks, we explored the semantic correlations between clock timestamps and scenes, and here we provide several examples to illustrate these connections. The Time Encoder and Image-Time Adaptor modules are designed to align visual CLIP representations and clock time embeddings. As a result, the learned time embeddings naturally align with CLIP text embeddings. This alignment allows us to factorise text concepts using TICL time embeddings and vice versa. Specifically, for each input text embedding, $T_{CLIP}$, we compute their similarity with time-class embeddings, $T_i$, using the Softmax function:

$$\textbf{Softmax} = \frac{\exp\left(T_{CLIP} \cdot T_i\right)}{\sum_{j=0}^{|C|-1} \exp\left(T_{CLIP} \cdot T_j\right)}$$

where $T_i, T_j$ are the TICL class embeddings. This formulation offers a probabilistic measure of the similarity between text embeddings and time classes. The resulting 24-hour class probabilities are shown in Fig. 35.

The results clearly demonstrate that texts describing specific times of day are directly associated with corresponding time periods. In addition, we also observe indirect associations. For example, the word "breakfast" is by definition related to morning hours, while "thief" is often associated with nighttime activities. These uneven probability distributions across the 24-hour timeline reflect the natural relations between certain events, scenes, or concepts and their corresponding time periods.

However, some irregular trends in the probability distributions indicate that our time-aware embeddings, learned from a limited image dataset, still have room for further improvements, particularly for night-time related concepts, which corresponds with fewer night-time image samples in the dataset. This highlights the need for further improvement of the dataset/model to achieve more robust time-awareness across all clock time periods.

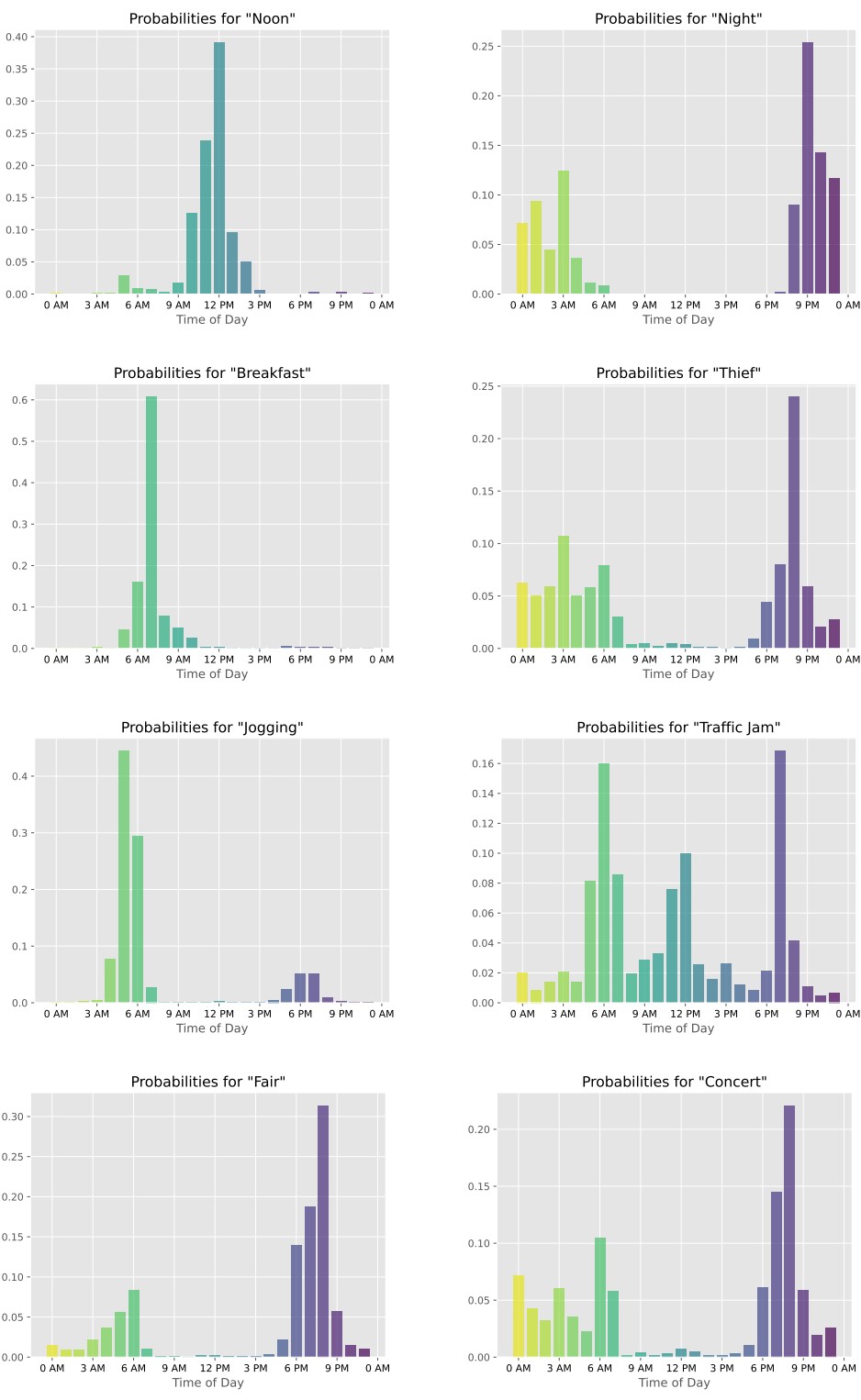

Figure 35: **Probability measure of the similarity between time classes and text queries.** The x-axis is hour classes and y-axis is probabilities calculated by **Softmax**.

