# OpenReview forum: "What Time Tells Us? An Explorative Study of Time Awareness Learned from Static Images"
_TMLR — Accepted by TMLR_

### Review · Reviewer_C7jQ · 2025-05-19

**Summary Of Contributions:**

This paper proposes a time-aware contrastive learning framework that aligns visual features with temporal representations for predicting the time coordinates from images. The paper also proposes a newly curated dataset for timestamp prediction.

**Audience:**

Yes

**Claims And Evidence:**

Yes

**Requested Changes:**

I would like to see more discussions regarding the dataset, particularly what makes the new dataset useful and how is it different than previous works.

**Strengths And Weaknesses:**

Strengths:

* The paper is well-motivated and easy to follow.
* The proposed model empirically performs well on several benchmarks.
* The experiments and ablation study are comprehensive. The paper ablates different vision backbone / image encoder and time coordinate transformations, and also conducts study on tasks like temporal aware image editing.
* The findings and analysis of why video scene classification can be improved via incorporating TICL are interesting.

Weakness:

* The proposed method is a combination of several well-established techniques (e.g. GeoClip), therefore the technical advancement could be limited. However, I think these combinations are well-motivated and supported by the ablation. The scope of the proposed method can be relatively limited as it only tackles hour prediction.
* The discussion of the newly proposed dataset is quite short for the main body part. A more thorough comparison with existing dataset (e.g. what is the main gap being addressed in this dataset) would help clarify its significance and utility for the hour prediction task.

---

> ### Author Response · Authors · 2025-06-21
> **Response to Reviewer C7jQ**
>
> Thank you for your thoughtful and careful review of our paper. We appreciate both your positive feedback and your constructive suggestions for improvement. Below, we respond to each of your comments and outline the revisions we made.
>
> ### **Limited Scope of the Method**
>
> > "The scope of the proposed method can be relatively limited as it only tackles hour prediction."
>
> Our approach is not limited to predicting time of day. In **Table 6**, we reported results on an additional task of **month-level date estimation** where our method outperforms all prior baselines. This suggested that the methodology could generalise to other transient attributes such as *season* and beyond.
> For the time-of-day task specifically, we chose 24 hourly bins as the main pre-text setting. **Appendix C.2** **(page 29-30)**  showed that increasing the resolution beyond 24 classes hurts robustness near class boundaries, a trade-off we discussed in the paper.
>
> ### **Main Requested Change – Expanded Discussion of the TOC Dataset**
>
> > The discussion of the newly proposed dataset is quite short for the main body part. A more thorough comparison with existing dataset would help clarify its significance and utility for the hour prediction task.
>
> We agree that the dataset contribution merits more space. Below, we first compare the proposed **TOC dataset** with prior timestamp-bearing datasets and then clarify the curation rationale. We have now included this to **Appendix A.1 (page 19-21)** in the revised version.
>
> | **Dataset** | **Image source** | **Timestamp reliability** | **Scene diversity** |
> | --- | --- | --- | --- |
> | MIRFLICKR-1M | mobile / miscellaneous | ✗ | high |
> | YFCC100M | mobile / miscellaneous | ✗ | high |
> | AMOS | fixed outdoor webcams | ✗ | limited |
> | CrossView Time (CVT) | webcams + mobile devices | ✗ | mixed |
> | **TOC (ours)** | **wild-view natural photography images** | **verified & timezone-aligned** | **high** |
>
> *Shortcomings of prior datasets:*
>
> - **MIRFLICKR-1M & YFCC100M** largely ignore timestamps; many images have missing or unsynchronised metadata.
> - **AMOS** offers accurate capture times but consists of static webcam views with potential **label leakage** (timestamps overlaid in the frame) and limited scene diversity (almost all of them are outdoor landscapes).
> - **CrossView Time (CVT)** inherits problems from both AMOS and YFCC100M: (i) many YFCC images are non-photographic (paintings, memes, etc.) and lack illumination cues; (ii) EXIF timestamps can be incorrect (early-2000s devices often lacked NTP synchronisation or used purely local clocks); and (iii) webcam images still risk visual label leakage where time can be overlaid on different places of the footage.
>
> In contrast, our cleaned dataset **TOC** has mitigated these problems through the two-stage cleaning pipeline (DBSCAN outlier removal → manual audit) described in Section 4 and Appendix A. It therefore contains image data containing various scenes (indoor/outdoor/landscape/portrait, etc.) and each of them are paired with timestamp metadata that are at least align with human perception. As a result, simply retraining the baseline network with the cleaned training split improves the performance of Salem et al. from 12.0 → 13.9 % (Top-1 Acc) on a natural-image test set (**Table 1, page 5**), illustrating that a cleaned TOC dataset can boost the pre-text task of timestamp prediction. We have updated the manuscript to provide more analysis and details on the impact and benefits of the new dataset.
>
> ### **Related updates**
>
> 1. **Add visualisations comparing sample-illumination distributions before and after our filtering process (in Appendix A.1, page 20, Figure 13)** It shows that, before cleansing, non-surveillance-camera images in the previous dataset taken during 00:00–06:00 share brightness levels similar to daytime samples, and there is a significant illumination gap between samples taken at 21:00–23:59 and 00:00–03:59. This trend contradicts common sense about the day–night cycle, in which illumination changes most dramatically around sunrise and sunset. The contradictory distribution results from poorly synchronised capture timestamps. After cleansing, the new dataset exhibits the expected natural trend. *We have included the figure and discussion in the revision to support the argument.*
>
> 2. **Include the dataset-comparison table with proper visualisations of previous issues (Appendix A.1, page 19, Figure 11 and Table 6)**, which highlight existing issues of e.g. poor visual quality, unsynchronised metadata, label leakage, etc. to clarify the necessity of the cleansing steps.
>
> We have updated the manuscript with changes highlighted in **blue** colour. We believe these changes will clarify both the scope of our method and the significance of the TOC dataset. Please let us know if you have any further questions and we'd be more than happy to clarify. Thank you again for your positive and insightful review that strengthens our paper.

---

### Review · Reviewer_vBRf · 2025-06-10

**Summary Of Contributions:**

This paper introduces a dataset and a machine learning method to identify the time at which the image was taken. The Time-Oriented Collection (TOC) dataset contains 130906 images, each of which has its associated timestamp of when it was taken. This is in itself a very valuable contribution of this work. The second contribution is Time-Image Contrastive Learning (TICL), a method by which we learn how to predict the time of the image. In all the paper is well written, and the ideas are clearly presented. The main issues that I identify are (i) the contribution of TICL, (ii) the importance and significance of the work, (iii) the quality of the numerical experiments.

**Audience:**

No

**Broader Impact Concerns:**

See before.

**Claims And Evidence:**

Yes

**Requested Changes:**

See before.

**Strengths And Weaknesses:**

Positive Points:
P1 - The paper is well written and the ideas are clearly presented. The figures are useful, and illustrative.

P2 - The dataset TOC seems like an important contribution to the field of machine learning. I believe that in the topic of understanding the capabilities of ML, it might be beneficial to assess how training an LLM on another task might correlate with properly predicting the time of the day for example. In any case, I believe that in itself, this dataset is an important and useful contribution.

Negative Points
N1 - It seems like previous works have already explored the idea of learning to predict the time of the day.
Examples are:

Anna Volokitin, Radu Timofte, and Luc Van Gool. Deep features or not: Temperature and time prediction in outdoor scenes. In 2016 IEEE Conference on Computer Vision and Pattern Recognition Workshops (CVPRW), pp. 1136–1144, 2016. doi: 10.1109/CVPRW.2016.145.


N2 - I believe that the contribution of TICL is minor if any. Essentially, a CLIP model with an extra learnable layer is used to align with time via contrastive learning. This is not a significant contribution to the field of ML. Essentially, this same idea could be used to do any other task: predict the age of the person in the image, classify the element in the image, etc. In other words, the way the problem is solved is super standard. Even more so, the “contribution” of the work is to introduce the contrastive learning approach, as opposed to a classifier with 24 classes (or any number of classes associated with any times of the day). I cannot understand why a simple classifier – that would remove the time econder – is not a better solution. Also, why is clip used instead of training a CNN from scratch.

N3 - The importance of the work is limited. Why would you want to train a model to predict the time of the day regardless of the position and the date? The variability is so high that it seems like a poorly posed problem. For most places sufficiently far away from the equator, the annual variability in sunlight is so large that very little if anything can be said about the time of the day regardless of the position and date.

N4 - The results are not interpretable. In the numerical values of Tables 2, it is very difficult to make sense of what those numbers even mean, given that the problem (as I said before) has a lot of noise. What is for example the human baseline? What is the performance of say ChatGPT or any other multimodal contrastive model?

N5 - The numerical experiments do not work very well. If we look at Figure 9, all of the underlying objects in the image change as time changes. In the bedroom for example, the sheet changes color. This clearly shows that the solution cannot be used for almost any real world application. This is a serious drawback for a paper that does not introduce any substantial theoretical or methodological ideas.

---

> ### Author Response · Authors · 2025-06-21
> **Response to Reviewer vBRf (part 1)**
>
> We thank the reviewer for the constructive feedback and for recognising the clarity of our writing (P1) and the potential contribution and importance of the TOC dataset (P2).
> Below we address each concern point-by-point, and have updated the revised version accordingly.
>
> ---
>
> ### **Response to N1 – “Prior work already predicts time-of-day”**
>
> > "It seems like previous works have already explored the idea of learning to predict the time of the day."
> >
>
> Firstly, we wish to reinstate that our major contribution is **not** solely on the time estimation pre-text task, but more on the pivotal question initially posed (and highlighted in the title): *"What time tells us from static images?"*  This makes our work significantly different to previous works that only focuses on accurate metadata (time, geolocation, temperature, etc.) estimation. Besides this major advancement, there are several additional differences makes our work stand out:
>
> - **Data scale & diversity** – TOC covers world-wide social-media photos with verified timestamps and various scenes, addressing the diversity and noise issues the prior works acknowledged.
> - **Independent to auxiliary metadata** – Unlike previous works, TICL needs *images only*, avoiding the brittle dependence on GPS/season and therefore generalising better to AMOS. And enabling downstream tasks for samples in which additional metadata other than timestamp is absent.
>
> We summarise the major differences in the table below, and these differences were discussed in the **Section 1 (page 1-3)** and **Section 2.2 (page 3)**:
>
> ---
>
> | **Data coverage ( ↔ work)** | **Additional metadata used?** | **Explicit time embedding?** | **What time tells us?** |
> | --- | --- | --- | --- |
> | **Volokitin et al., 2016** – 6 fixed webcams, daylight (10h) only | Temperature | ✗ | ✗ |
> | **Zhai et al., 2019** – Mixed Flickr + webcam set, 24 h × 12 months → 288 classes | GPS Coordinates | ✗ | ✗ |
> | **Salem et al., 2022** – Large webcam corpus, year-round, day & night | GPS Coordinates | ✗ | ✗ |
> | **This work (TICL)** – 130 k social-media photos worldwide, day & night | ✗ | ✓ | Reported via downstream tasks |
>
> ---
>
> ### **Response to N2 – “TICL is just CLIP + a layer, lacking justification”**
>
> > " I believe that the contribution of TICL is minor if any."
> >
>
> Again, we wish to clarify that TICL’s contribution is conceptual rather than architectural. Our focus is **not** to provide a highly specialised network perfectly optimised for the pre-text time-estimation task. Instead, our goal is to explore *“what time tells us”* with a simple, scalable framework that is extensible to different tasks. Contrastive learning and a pretrained CLIP model are convenient tools for this purpose. We have included more discussion in **Section 3 (page 3-5)** for clearer contribution statements.
>
> > “Essentially, this same idea could be used to do any other task…"
> >
> - We agree that the same idea could be applied to other attributes, such as age estimation. However, we want to emphasise that such potential underscores its generic value rather than diminishing its contribution.
>
> > "I cannot understand why a simple classifier – that would remove the time econder – is not a better solution. "
> >
> - In TICL, time classes are embedded free vectors, not fixed one-hots, so neighbouring hours naturally share information. *An explicit embedding produced by the time encoder not only facilitates analysis of the learned semantics and enables downstream tasks that require only timestamp inputs, but also improves pretext performance.* These class embeddings, which are made possible by the time encoding module, also help us investigate what the time awareness learned from visual data semantically represents textual concepts in **Figures 7 (page 11)**, and **Figure 35 (page 40) (originally Figure 32)**, revealing both direct ("breakfast", "lunch") and indirect ("concert", "thief") associations.
> - In addition to the above benefit, our ablation (**Table 3**) also shows that *either* the Image Encoder *or* the Image-Time Adaptor alone (a simpler classifier design) is insufficient; only their combination yields the full gain across four backbones.
>
> > "Also, why is clip used instead of training a CNN from scratch."
> >
> - We chose a frozen CLIP backbone with a lightweight adaptor instead of training a CNN from scratch. **Section 3 (Page 3-5)** already explained this design choice. Previous works that train their own backbones, with or without additional metadata, show limited performances and biased predictions (**Figure 4 at page 7** and **Table 2 at page 8**).
>
> (Continued...)

---

> ### Author Response · Authors · 2025-06-21
> **Response to Reviewer vBRf (part 2)**
>
> ### **Response to N3 – “Limited importance of time-of-day prediction”**
>
> > "The importance of the work is limited."
> >
>
> We understand the concern that predicting the capture time of natural images *without* GPS/date may seem noisy. However, it has practical applications, including:
>
> 1. **Media forensics**: detecting manipulated EXIF data (Padilha et al., 2022 [1]; cited in **Section 2.1, page 3**).
> 2. **Retrieval & recommendation** – our experiment shows users can query archives by *time context* more accurately with the proposed approach, with potential usage of ordering unlabelled photos/videos **(Section 6.1, page 9 and Appendix D, page 30)**.
> 3. **Scene-understanding priors** – time cues improve dynamic-scene classification (**Table 5, page 11**), confirming that time can lead the model to learn useful scene features.
>
> > "Why would you want to train a model to predict the time of the day regardless of the position and the date?"
> >
>
> The motivation of the pretext task in which we train a model to predict the time of the day is to study how static visual appearances correlate with the concept of time-of-day. I.e., pre-text timestamp estimation itself is not our primary interest; rather, we focus on the overarching question *“What does time tell us?”* in computer-vision tasks, verified through multiple downstream experiments. In this case, we wish to get purely visual embeddings only about time, and get rid of the dependency on the additional metadata as they are independent to timestamps. Despite these factors like geolocation/climate result in uncertainties of illumination appearances for certain clock times, such uncertainties also reflect how our visual impressions about clock time are shaped.
>
> Our exploratory findings suggest that time (1) accounts not only for natural-light illumination (generative editing in **Section 6.3, page 10-12** and **Appendix F, page 34-39**) but also provides (2) direct and indirect semantic relations in text/video. Consecutive video frames representing short periods share similar time-aware features even with drastic visual changes (retrieval in **Section 6.1, page 9**), supplying a strong prior for per-video scene classification (**Section 6.2, page 9-10**). We believe these insights will interest at least some of the TMLR audiences, and potentially to a larger group in the Computer Vision community.
>
> ---
>
> ### **Response to N4 – “Results not interpretable”**
>
> > "The results are not interpretable."
> >
>
> We understand the reviewer's concern that the  the metrics can be hard to parse, though they are standard in prior works. To address this concern, as suggested, we 1) conduct a user study to provide an estimate for human performance on the task. and 2) conducted an additional experiment with VQA models including ChatGPT (specifically GPT-4o-mini) and other multi-modal models which provides a baseline performance other than previous works solely focusing on attribute estimation tasks.
>
> Due to the time limit, we are not able to conduct a complete user study over all the 13k + 3k test samples. The suggested additional experiment about evaluating human capability of clock time are therefore based on 45 randomly selected images, each of them was shown to 11 human participants, to perform a user study. In this study, each user was asked to answer a question of “Please indicate your estimation of captured clock time of the image shown below”. This user study therefore provide an estimation of human capability on the task. The results are shown in the table below:
>
> On TOC test set:
>
> | **Model** | **Top-1 Acc. (%) ↑** | **Time MAE (min) ↓** |
> | --- | --- | --- |
> | *Human Baseline* | 11.73 | 228.21 |
> | BLIP-VQA-base  | 9.36 | 241.58 |
> | GPT-4o-mini | 15.35 | 161.39 |
>
> On AMOS test set:
>
> | **Model** | **Top-1 Acc. (%) ↑** | **Time MAE (min) ↓** |
> | --- | --- | --- |
> | *Human Baseline* | 11.52 | 235.01 |
> | BLIP-VQA-base | 7.28 | 302.82 |
> | GPT-4o-mini  | 11.15 | 216.82 |
>
> Note that for all experiments under VQA-based pipelines, we are unable to report the top-3 and top-5 accuracies as the VQA paradigm does not provide additional prediction candidates. From the results, it can be seen that our approach still performs better than the suggested additional methods (i.e. ChatGPT and humans).
>
> We hope including the full testing VQA results to **Table 2** could improve the clarity of the quantitative performances on the pre-text task, *though the improvements on benchmarks of pretext task are not our primary objective/focus nor the only contribution.*
>
> (Continued...)

---

> ### Author Response · Authors · 2025-06-21
> **Response to Reviewer vBRf (part 3)**
>
> ### **Response to N5 – “Limited qualitative results ⇒ method useless”**
>
> > "The numerical experiments do not work very well."
> >
>
> Please note that generative image-editing experiments are included **not** as a new application but as a sanity check: does the learned time-aware representation reflect plausible visual appearances? Despite some results in **Figure 9 (page 12)** may show limited visual quality, the qualitative and quantitative gains with the time-aware loss guidance compared to the baselines still supports our assumption.
>
> Besides, the limitations in **Figure 9** stem from the baseline editing methods we adopted (Styleclip [2] with base model StyleGAN-v2), as can be seen from the corresponding results from those methods as comparison. We chose these baselines because they minimise dependencies on other multimodal models, letting us better isolate the impact of the time-aware loss. Our time-aware loss is plug-and-play and can be applied to stronger image editing backbones, as shown in **Appendix F (page 34-39)**, where diffusion-based backbone models yield better visuals (**Figures 32-34**, which are originally **Figures 29-31** in the initial submission).
> Again, please note we **never claim** our method to be an image synthesis/editing work, and our aim for this experiment was **not** achieving a high-quality visual appearance nor "un-change objects", but rather on the effectiveness of our learned time-aware representations, as highlighted throughout our paper.
>
> Finally, indoor day/night changes are naturally subtle, as the illuminations are dominated by artificial light sources (mostly invariant subject to time of day!) if any. LSUN-Bedroom samples therefore serve as extreme failure cases, where our method nevertheless slightly outperforms the baselines.
>
> ---
>
> ### **Requested Revisions**
>
> | **Reviewer request** | **Updates we take** |
> | --- | --- |
> | Explain model design contributions (N2) | Already in **Section 3 (page 3-5)**, we have updated the wording to focus on the conceptual gain of the model design |
> | Clarify application importance (N3) | Expanded **Section 6 (page 9-12)** with a discussion on explicit application taxonomy (forensics, retrieval, etc.) |
> | Make results more interpretable (N4) | Added additional baseline results to **Table 2 (page 8)** |
>
> ---
>
> We appreciate the reviewer’s careful suggestions and believe the updates with regards to the suggestions improve the clarity and quality of the paper.
>
> We have updated the manuscript with changes coloured in blue . We hope the revision and response above addressed the concerns raised and strengthen the paper. We would be more than happy to clarify further if the reviewer has any other questions.
>
> ---
>
> [1] Padilha, Rafael, et al. "Content-aware detection of temporal metadata manipulation." *IEEE Transactions on Information Forensics and Security* 17 (2022): 1316-1327.
>
> [2] Patashnik, Or, et al. "Styleclip: Text-driven manipulation of stylegan imagery." *Proceedings of the IEEE/CVF international conference on computer vision*. 2021.

---

> > ### Comment · Reviewer_vBRf · 2025-06-27
> > **Reply**
> >
> > I would like to thank the authors for their responses.
> > My comments and concerns were properly taken care of.
> > I am not fully satisfied with the answer from the authors. The "mantra" that the authors keep reiterating on is that the purpose of their work is to "our goal is to explore “what time tells us” ". However, upon reading the paper, very little is provided as an answer.
> > I fail to understand how a simple implementation on top of CLIP can actually tell us anything about time.
> > I appreciate the fact that the authors have added work to support their paper.

---

> > > ### Author Response · Authors · 2025-06-28
> > > **Reply to Reviewer vBRf**
> > >
> > > We thank the reviewer for their reply and their appreciation of our additionally provided supporting information. Regarding the remaining unsatisfactory answer, we emphasised our aim (i.e. what time tells us from a static image) was to remind the main contribution of our work, and tried to avoid any potential misunderstanding that the primary contribution of our work is related to network architectures.
> > >
> > > Here “what time tells us” intends to say, by predicting the time of the day from a given static image, what we can learn from it.
> > >
> > > To answer the question of “what is learned about time”, we presented a study over a few downstream tasks, as shown in Section 6, in which we showed how the learned representation could tell us properties about time. For example, it can help retrieve images with similar captured time; the learned time-aware representations can help video scene classification; it can also provide time-related guidance for image editing. Additionally, we showed visualisations of how the learned representations/embeddings are related to time in the main paper and the appendix.
> > >
> > > We hope this further clarifies it and hope the remaining concern is now addressed. But we are more than happy to engage in further discussion if the reviewer has any further concerns.

---

### Review · Reviewer_VzWe · 2025-06-12

**Summary Of Contributions:**

For the task of estimating time a photo is taken, the authors collect a dataset of 130,906 images with timestamps (and GPS coordinates). To address the task, the authors propose a contrastive learning objective to co-embed a CLIP embedding and a time vector into a single embedding space. Using the proposed dataset as the training set, the authors are able to achieve better test accuracy both on TOC and AMOS test sets. The authors show detailed experimental results by replacing the CLIP encoder with other encoders. By reducing the time estimation error, the method shows lower geolocation estimation error, higher video scene classification accuracy, time-aware image editing (Zero shot setting).

**Audience:**

Yes

**Claims And Evidence:**

Yes

**Requested Changes:**

- works -> work
- e.g. -> e.g.,
- "Similarly, Salem et al. (2022)" are marked as a hyperlink while "Salem et al. (2022)" only should be marked as a hyperlink.
- Ablation study -> detailed study: The ablation study means that you need to remove each (proposed) component and observe the performance drop due to the absence of the proposed components. Here, you replace the CLIP encoder with other encoders to observe the performance difference. This is not an ablation study.

**Strengths And Weaknesses:**

**Strengths**
- Method is simple and works for a number of different tasks (e.g., time based image retrieval, geolocation estimation error, video scene classification, time-aware image editing)

**Weaknesses**
- The proposed method is too simple and technically not novel.
	- All components (contrastive loss, CLIP embedding) are very popular for co-embedding of data in two modalities.
- In addition to Fig. 6, it is better to see the quantitative comparison of errors by a number as it is not clear to see the benefit of the method.
- Performance of Time-aware image editing is hard to measure because it only shows a few qualitative examples.
- Although it is clear that performance of time-based image retrieval is improved by better time-image co-embedding, it is not clear that the performance gain of other zero shot task. The paper has no good discussion on it.
- Main text is not self-contained. A number of unclear presentation is clarified in appendix without pointer.
	- Cleaned TOC dataset (looks like A.2 is explaining the cleaning procedure)
	- In Figure 4, there is TICL-NN. But it is never explained in the main text (but in the appendix). The main text should have a pointer to the appendix explaining the TICL-NN.

---

> ### Author Response · Authors · 2025-06-21
> **Response to Reviewer VzWe (part 1)**
>
> We thank the reviewer for the comments and  feedback. However, we wish to firstly clarify the misunderstanding before detailed response to each of the points mentioned.
>
> > "...the method shows lower geolocation estimation error... "
>
> - We **never** mentioned that our method shows lower geolocation estimation error. We did **not even try this task (image localisation/geolocation estimation)**. To clarify, we only looked at the mean geolocation errors in the top-100 retrieved samples of the time-based image retrieval task in **Figure 6 (page 10).** From the result, we noticed that, while the TICL model shows smaller mean time error in retrieved samples, the mean geolocation error is slightly larger compared with raw CLIP model but lower than other purely time estimation models. This inspired an attempt to the new task of joint localisation of geolocation and time (**Table 4, page 9**), which can be a useful tool for image forensics and several applications. Our method have reached better performance than other baselines. All these discussions are originally provided in the second paragraph of **Section 6.1 (page 9)**.
>
> ### **Response to Weaknesses**
>
> > "The proposed method is too simple and technically not novel."
>
> - **Please note that, as highlighted in our title and throughout the paper, the primary focus of this work is to answer the question "*What time tells us*", rather than build a fancy specialised model for the time estimation task. We also did not claim any such novelty.** Therefore, the contribution of our paper is conceptual rather than architectural, which is, to our knowledge, the first to study the correlation of the captured time of input data to other computer vision tasks.
>
> > "... it is better to see the quantitative comparison of errors ..."
>
> - We thank the reviewer’s suggestion. As suggested, we have updated **Figure 6, page 10** with numerical values of mean errors across the top-100 retrieved samples, in the revised version.
>
> > "Performance of Time-aware image editing is hard to measure because it only shows a few qualitative example"
>
> - Due to space limit, we were unable to provide all the time-aware image editing results in the main paper, however, we have already provided necessary quantitative results for measuring image editing performances in the Appendix, specifically FID scores in **Table 11 at page 36 (originally Table 10)**, User study in **Table 12 at page 37 (originally Table 11)**, along with more quantitative examples in **Figure 29-34 (originally Figure 26-31)**. These results are accompanied with discussions in **Appendix F (page 34-39)**, which we provided pointers in **Section 6.3 (page 10-12)** in the main paper. We have updated the manuscript with more fine-grained pointers.
>
> > "Although it is clear that performance of time-based image retrieval is improved by better time-image co-embedding, it is not clear that the performance gain of other zero shot task. The paper has no good discussion on it."
>
> - In the original submission, we already covered another zero-shot task of time-aware image-editing in **Section 6.3, page 10-12** and **Appendix F, page 34-39**, in which the editing process is essentially latent optimisation without need for any training process. We hope the performance gain on this zero-shot task reveals another aspect to evaluate the possible value of learned time-aware visual embedding.
>
> > "A number of unclear presentation is clarified in appendix without pointer."
>
> - For "Cleaned TOC dataset" mentioned in **Section 3 (page 3-5)**, we have now updated it with a pointer to **Appendix A.2 (page 21)**.
> - TICL-NN has a simple in-place description in the footnote of the **Table 2 (page 8)**, with the pointers in the end of **Section 4 (page 6).**  We have now further specified the pointer to the corresponding footnote and the main text sentences linking to **Appendix B2, page 22**.
>
> (..continued)

---

> ### Author Response · Authors · 2025-06-21
> **Response to Reviewer VzWe (part 2)**
>
> ### **Requested Changes**
>
> > "works -> work..."
> >
> - Thank you for pointing out this. We have now corrected them in the revised version.
> - “e.g.” is also a correct representation in English. To be consistent, we use “e.g.” throughout the paper.
>
> > "Similarly, Salem et al. (2022)" are marked as a hyperlink while "Salem et al. (2022)" only should be marked as a hyperlink.
> >
> - Could you please point us to this specific formatting error? The only place we can locate "Similarly, Salem et al. (2022)" is on (the originally submitted version) page 3, 2nd last paragraph, L8. But the hyperlink was only applied to "Salem et al. (2022)".
>
> > "Ablation study -> detailed study..."
> >
> - As suggested, we updated the title of **Section 5.3** and **Table 3 (page 7-8)** from *"Ablation study"* to *”Detailed component analysis“*, while we wish to clarify that **Section 5.3** and **Table 3** originally contains ablation study by reporting performances when removing part or all of the proposed components for a suite of image encoder we tested.
>
> ---
>
> In summary, we thank the reviewer for their valuable comments helping us to improve the paper. We have updated the manuscript accordingly with changes coloured in blue. We hope the revision and response have now addressed the concerns raised and strengthen the paper. We would be happy to answer further questions if the reviewer still has.

---

### Review · Reviewer_uEWw · 2025-06-14

**Summary Of Contributions:**

This paper studies time-awared representation learning.  Capturing time from input images is not merely detecting illumination changes from input images.  While varying time induces illumination changes in the visual scene, other factors such as weather, geolocation and season convolute the appearance of observed images.  This paper explores if existing performant image encoders, capable of distinguishing semantics, can also extracts time-of-day awareness.  The contribution of this paper is three-fold: (1) It curates a new benchmark dataset with reliable timestamps across a high variety of geolocations, (2) It proposes a time-image contrastive learning framework for extracting distinctive time-awared features from static images, and (3) It provides multiple examples of downstream tasks, demonstrating the application value of the studied problem.

**Audience:**

Yes

**Claims And Evidence:**

Yes

**Requested Changes:**

1. I'd encourage the authors to compare against Segsort--a supervised contrastive learning alternative.  Probably the authors can obtain more striking results with the relaxed contrastive learning.

**Strengths And Weaknesses:**

Streghts:

1. The idea of studying time-awared representation learning is interesting.  When reading the introduction, I had concern about the application value of the studied problem, which was clearly addressed in Section 6.  Esepcially, the application on time-aware image editing is impressive.

2. This paper provides strong argument on the choice of learning framework.  It clearly illustrates the issues of regression-based methods, which often predict midpoint of the range, and the problems of classification-based methods, which overlook the structure between different timestamps.  Based on this observation, the paper proposes a contrastive learning framework, that leverages a strong pre-trained image encoder as prior and learns to distinguish images captured at different time.  The reasoning is technically sound and supported by the experimental results.

3. This paper provides thorough analyses to validate the efficacy of the proposed method, including confusion matrices, time estimation performance, time-based image retrieval, video classification, joint localisation of geolocation and time, t-SNE visualisation comparison, and time-aware image editing.  I'm convinced by all these results.

4. The authors put much effort in presenting different downstream tasks.  I'm convinced that the studied research problem indeed has real-world application value.

5. The paper is clearly written.  Readers can easily grab the high-level idea, implementation details and conclusion of experimental results.

---

Weaknesses:

1. My only concern is that time-image contrastive learning may not be the most optimal choice of time-aware learning framework.  Since all images captured at the same time are grouped to the same timestamp prototype, it overlooks that the same "clock time" has different physical meaning at different locations or seasons.  For example, 5 am may be considered late night during winter but early morning during summer.  The proposed method learns to predict similar features of images captures at 5 am in summer and winter, which definitely will sabotages the efficacy on downstream tasks (e.g. time-aware image editing).

An obvious alternative is supervised contrastive learning [1] that groups and separates images of similar and different categories in the latent feature space, rather than mapping images to the corresponding protype features.  Such relaxed contrastive learning may help the model to respect the appearance of input images and to distinguish images captured at different time, geolocation, season and weather.

2. Following Weakness 1, an interesting idea is to normalize timestamps in different seasons.  For example, what time in winter would 5 am in summer correspond to?  Such results may help the scientific community to calibrate bio-clock of human beings.

---

Reference

[1] Hwang, Jyh-Jing, et al. "Segsort: Segmentation by discriminative sorting of segments." Proceedings of the IEEE/CVF International Conference on Computer Vision. 2019.

---

> ### Author Response · Authors · 2025-06-21
> **Response to Reviewer uEWw (part 1)**
>
> We sincerely thank Reviewer uEWw for the thoughtful and encouraging feedback. Below we address the weaknesses and the concerns point by point.
>
> ### **Weakness 1: Exploration on Segsort-style supervised contrastive learning**
>
> > “I'd encourage the authors to compare against Segsort-a supervised contrastive learning alternative”
> >
>
> Thanks for the insightful advice, we have tried the idea of using a relaxed supervised contrastive learning by following constructions:
>
> 1. Changing the original InfoNCE style contrastive learning to the Segsort style [1]. The model repeatedly clusters each hour’s embeddings with K-means and then pulls together those images that share the *same* hour-specific sub-cluster in a minibatch (memory bank) while pushing all others using the negative log-likelihood loss.
> 2. With the learnt model, during inference, we embed a new image once and predict its clock time by doing a cosine *k-nearest-neighbour* search against the frozen gallery of train embeddings.
> 3. The settings of additional hyperparameters (we followed original settings of Segsort) used are listed below (other hyperparameters are set to the same as the original TICL):
>
> | **Hyperparameter name** | **values** |
> | --- | --- |
> | `n_sub` | 25 |
> | `max_iter` | 10 |
> | `concentration_constant` | 10 |
>
> On TOC test set:
>
> | **Methods** | **Top-1 Acc (%) ↑** | **Top-3 Acc (%) ↑** | **Top-5 Acc (%) ↑** | **TimeMAE (min) ↓** |
> | --- | --- | --- | --- | --- |
> | Segsort-style | 18.97 | 39.32 | 52.27 | 171.55 |
> | InfoNCE (Classification) | 20.60 | 49.01 | **67.82** | 171.65 |
> | InfoNCE (Nearest-neighbour) | **25.67** | **49.32** | 66.74 | **156.24** |
>
> On AMOS test set:
>
> | **Methods** | **Top-1 Acc (%) ↑** | **Top-3 Acc (%) ↑** | **Top-5 Acc (%) ↑** | **TimeMAE (min) ↓** |
> | --- | --- | --- | --- | --- |
> | Segsort-style | 12.29 | 28.37 | 39.57 | 209.00 |
> | InfoNCE (Classification) | **13.55** | **38.49** | **57.28** | **187.87** |
> | InfoNCE (Nearest-neighbour) | 11.14 | 31.01 | 48.84 | 220.94 |
>
> We noticed that by directly applying this idea, the performance is not consistently improved, possibly due to the following reasons:
>
> - In Segsort, the negative log-likelihood loss try to segment each pixel to different semantic prototypes, each represents possible object classes. However, in our problem setting, the difference between different times may not be as distinguishable as visual differences between classes.
> - Also, since we are trying to discover sub-modes within each time periods using k-means over CLIP features, it is likely to reach to a degenerated solution where the sub-modes simply represents different main objects which makes the semantics a lot more different in the early training iterations. While such clusters may be good for distinguishing objects, they may not provide useful guidance for us to discover different sub-modes about clock time.
>
> As a result, while we acknowledge the suggested segsort-style supervised contrastive learning is an interesting idea, it may need further in-depth investigation in future (a bit out of the scope of this paper) to find the best way to leverage the time-related supervision signal for supervised contrastive learning.
>
> (... continued)

---

> ### Author Response · Authors · 2025-06-21
> **Response to Reviewer uEWw (part 2)**
>
> ### **Weakness 2: Normalising Clock Time**
>
> > “…an interesting idea is to normalize timestamps in different seasons…”
> >
>
> Thanks for the suggestion, this is an interesting question! Unfortunately, the illuminations and appearances of different time of day can be dominated by many other factors other than clock time on earth (It would be ideal if the clock time can be isolated).  Therefore, in this work we may not be able to propose an absolutely accurate mechanism to normalise the clock time with regards to season/climate/date/geolocation, etc.
>
> Despite such complexity, to further explore this interesting suggestion, we have designed a new experiment that we adjust our TICL model learn to jointly predict hour of day + season using a similar idea in [2]. Since the season is dependent on date and geolocation (different hemisphere!), it serves as a rough combination between the geolocation metadata and clock time. With this additional supervision, we were able to reach similar time estimation performance to purely clock time supervised models with a much smaller gap than combining clock time with other single metadata supervision such as date. That means, learning combinations about other metadata could give a better clue to normalise the visual variance of the clock time. We have updated the quantitative results to the **Table 7**, with the discussions in **Appendix B.4 (page 23-24), in the revised version**.
>
> - Clock time estimation:
>
>
>     | **Method** | **Top-1 acc ↑** | **Top-3 acc ↑** | **Top-5 acc ↑** | **Time MAE (min.) ↓** |
>     | --- | --- | --- | --- | --- |
>     | TICL (Season, Hour) | 20.14% | 45.98% | 62.58% | 170.93 |
> - Season estimation:
>
>
>     | **Method** | **Top-1 acc ↑** | **Top-3 acc ↑** | **Top-5 acc ↑** |
>     | --- | --- | --- | --- |
>     | TICL (Season, Hour) | 61.52% | 71.74% | 80.48% |
>
> After we learn such a model, we were able to visualize how time/season pairs (4*24 = 96 combinatorics) are placed in the representation space, that possibly, gives us clues of how clock time varies or shares similarities across different season. The visualization is now included in the revised version as **Figure 18 (page 23)**. The clear patterns within it possibly provide clues to calibrate the unnormalised clock time with regards to season. Further study may have the potential to develop a calibration mechanism converting the clock time to bio-clock through the joint supervisions across different metadata on visual samples.
>
> ---
>
> In summary, we would like to thank the reviewer again for the constructive suggestions and comments, which indeed helped us to improve the paper. We have updated the paper with all the changes highlighted in blue in the revised version.
>
> In the meantime, we'd be more than happy to clarify any further questions the reviewer might have.
>
> ---
>
> [1] Hwang, Jyh-Jing, et al. "Segsort: Segmentation by discriminative sorting of segments." Proceedings of the IEEE/CVF International Conference on Computer Vision. 2019.
>
> [2] Zhai, Menghua, et al. "Learning geo-temporal image features." *arXiv preprint arXiv:1909.07499* (2019).

---

### Decision · Action_Editor_mQ8u · 2025-09-03

**Recommendation:** Accept with minor revision

**Additional Comments:**

The majority of reviewers (C7jQ and VzWe leaning accept, with uEWw strongly positive) support acceptance, outweighing the single leaning reject from vBRf. The paper meets TMLR's criteria for soundness and audience interest, with claims supported by evidences. Minor revisions should address any remaining clarifications on novelty and methodological comparisons raised in rebuttals (e.g., supervised contrastive learning alternatives), and incorporating new results introduced in the rebuttal period to the manuscript.

**Audience:**

Yes

**Audience Explanation:**

All three official reviewers (C7jQ, VzWe, vBRf) unanimously answered "Yes" to this question. Reviewer uEWw highlighted the paper's interest in exploring time-aware representation learning and its applications, such as image editing and video classification. The topic of learning time-related visual cues from static images has close relevance to the machine learning community, particularly in tasks that require visual understanding capability. While the architecture/model is not novel, the task itself introduces novel aspects for training and evaluating multi-modal foundation models, and thus deemed valuable for TMLR's audience.

**Claims And Evidence:**

Yes

**Claims Explanation:**

All three official reviewers (C7jQ, VzWe, vBRf) unanimously answered "Yes" to this question in their recommendations. The additional review from uEWw also affirms that the claims are supported by thorough analyses, including confusion matrices, performance metrics, visualizations, and downstream task results. The authors' rebuttals further addressed minor concerns raised, providing additional experiments and clarifications that validate the evidence.